EMBO
Molecular Medicine

# Transcriptional response to hepatitis C virus infection and interferon-alpha treatment in the human liver

Tujana Boldanova[1,2], Aleksei Suslov[1], Markus H Heim[1,2,*,‡] & Anamaria Necsulea[3,**,†,‡] 🆔

## Abstract

**Hepatitis C virus (HCV) is widely used to investigate host–virus interactions. Cellular responses to HCV infection have been extensively studied *in vitro*. However, in human liver, interferon (IFN)-stimulated gene expression can mask direct transcriptional responses to infection. To better characterize the direct effects of HCV infection *in vivo*, we analyze the transcriptomes of HCV-infected patients lacking an activated endogenous IFN system. We show that expression changes observed in these patients predominantly reflect immune cell infiltrates rather than cell-intrinsic pathways. We also investigate the transcriptomes of patients with endogenous IFN activation, which paradoxically cannot eradicate viral infection. We find that most IFN-stimulated genes are induced by both recombinant IFN therapy and the endogenous IFN system, but with lower induction levels in the latter, indicating that the innate immune response in chronic hepatitis C is too weak to clear the virus. We show that coding and non-coding transcripts have different expression dynamics following IFN treatment. Several microRNA primary transcripts, including that of miR-122, are significantly down-regulated in response to IFN treatment, suggesting a new mechanism for IFN-induced expression fine-tuning.**

**Keywords** HCV; hepatitis C; interferon; miRNAs; transcriptome
**Subject Categories** Chromatin, Epigenetics, Genomics & Functional Genomics; Microbiology, Virology & Host Pathogen Interaction

## Introduction

Hepatitis C virus (HCV) infections are a major cause of liver-related morbidity and mortality. An estimated 160 million persons are chronically infected worldwide and are at risk to develop liver cirrhosis and hepatocellular carcinoma (Lavanchy, 2011). Because of their substantial impact on human health, HCV infections have been extensively studied. HCV is now one of the most widely used model systems to investigate host–virus interactions (Colpitts *et al*, 2015). HCV is transmitted through blood and infects and replicates in hepatocytes. Due to the lack of a small animal model and of the difficulties inherent to working with human samples, HCV–host cell interactions have been mainly studied in a cell culture system, specifically in Huh7-derived hepatoma cells infected with the JFH1 isolate of the virus (Lindenbach *et al*, 2005; Wakita *et al*, 2005; Zhong *et al*, 2005; Walters *et al*, 2009; Colpitts *et al*, 2015). Experiments in this *in vitro* system have identified a large number of host factors that are required for viral replication or that have antiviral properties (Colpitts *et al*, 2015). This experimental system also brought important insights into the virus–host interactions that may contribute to pathogenesis, for example, revealing cell cycle perturbations in HCV-infected cells (Walters *et al*, 2009). However, few of these findings have been evaluated in the human liver.

Studying HCV infections *in vivo* presents important challenges. An intrinsic difficulty comes from the immune response, which is a strong confounding factor in analyses of human liver biopsies. Gene expression differences between HCV-infected and uninfected livers are the result of direct HCV-induced cell-autonomous adaptive responses in infected cells and of more global changes that result from the immune response in the liver. The chronic phase of HCV infections is characterized by a largely ineffective cellular immune response combined with a highly variable interferon-lambda-mediated innate immune response (Heim & Thimme, 2014). A significant proportion of patients are characterized by an endogenous activation of the interferon (IFN) system, in which hundreds of classical IFN-stimulated genes (ISGs) are strongly induced (Heim & Thimme, 2014). The presence of the endogenous IFN system activation can mask more subtle changes that occur as a direct consequence of viral infection and replication in HCV-infected cells. The confounding effect of the immune answer is aggravated by the fact that the percentage of HCV-infected hepatocytes rarely exceeds 50%

1 Department of Biomedicine, University of Basel, Basel, Switzerland
2 Division of Gastroenterology and Hepatology, University Hospital Basel, University of Basel, Basel, Switzerland
3 Laboratory of Developmental Genomics, School of Life Sciences, École Polytechnique Fédérale de Lausanne, Lausanne, Switzerland
 *Corresponding author. Tel: +41 61 265 25 25; E-mail: markus.heim@unibas.ch
 **Corresponding author. Tel: +33 4 72 44 81 42; E-mail: anamaria.necsulea@gmail.com
 †Present address: Laboratoire de Biométrie et Biologie Evolutive, Université Lyon 1, CNRS, UMR5558, Université de Lyon, Villeurbanne, France
 ‡These authors contributed equally to this work

and often is below 20%, whereas ISG expression can be observed in up to 95% of cells (Wieland *et al*, 2014). To better understand the molecular consequences of HCV infection *in vivo*, it is thus important to disentangle the direct cellular response to viral infections from the transcriptional signature of the immune response, and in particular of the endogenous IFN system activation.

The endogenous activity of the IFN system is also highly relevant for therapeutic choice in chronic hepatitis C (CHC). Until the recent introduction of direct antiviral drugs for the treatment of CHC, recombinant pegylated IFN-alpha 2 (pegIFN-α) had been an essential component of the standard of care for CHC for over 25 years, and it is still used in many parts of the world. Treatment with pegIFN-α and ribavirin achieved cure rates between 30 and 80%, depending on the viral genotype, pre-treatment patient history, and stage of liver fibrosis (Heim, 2013). The success of the treatment is also highly dependent on the genetic background of the patients. Genome-wide association studies revealed significant associations between polymorphisms in the *IFNL4* gene and response to pegIFN-α/ribavirin (Bibert *et al*, 2013; Prokunina-Olsson *et al*, 2013; Terczynska-Dyla *et al*, 2014). The recently discovered IFNL4 protein has strong antiviral properties and stimulates ISG production through binding to the IFN-lambda receptor (Hamming *et al*, 2013; Prokunina-Olsson *et al*, 2013). The *IFNL4* gene harbors several genetic variants in human populations, including a frameshift mutation that abrogates the production of the IFNL4 protein (Terczynska-Dyla *et al*, 2014). Paradoxically, the IFNL4-producing genotype is associated with poor response to pegIFN-α/ribavirin, whereas mutated alleles coding for an IFNL4 variant with strongly reduced biological activity or even a complete loss of function are associated with very good spontaneous and treatment-induced resolution rates (Terczynska-Dyla *et al*, 2014). These observations are consistent with earlier findings that patients who have a strong endogenous induction of ISGs during the chronic phase of HCV infection do not respond to therapeutically injected pegIFN-α (Chen *et al*, 2005; Asselah *et al*, 2008; Sarasin-Filipowicz *et al*, 2008). It is presently not known why the activation of the endogenous IFN system in the liver in patients with the IFNL4-producing genotype is ineffective against HCV, whereas pegIFN-α-induced ISG expression results in viral eradication in so many patients.

In this study, we aimed to disentangle the direct and indirect effects of HCV infection on gene expression patterns, by performing a detailed characterization of the gene expression changes associated with HCV infection, endogenous IFN system activation, and pegIFN-α treatment in the human liver. With this objective, we generated and analyzed high-throughput transcriptome sequencing profiles from liver biopsies derived from different categories of HCV-infected and non-infected patients, prior to and during treatment. First, to unveil HCV-induced cell-autonomous effects and to separate them from IFN-induced changes in the transcriptome, we selected liver biopsies from CHC patients without hepatic ISG induction, and compared them with uninfected control biopsies. Second, we examined the transcriptomic changes associated with the endogenous activation of the IFN system in a subset of CHC patients. Finally, we analyzed the gene expression changes resulting from pegIFN-α/ribavirin treatment, by comparing transcriptome data from liver biopsies obtained before treatment and at different time points during the first week of therapy. We found that the transcriptional profiles associated with endogenous IFN activation and

with pegIFN-α/ribavirin treatment share a core set of IFN-stimulated genes, although quantitative differences can be found in gene activation levels.

Throughout our study, we investigated the differential expression patterns of both protein-coding genes and non-coding RNAs, aiming to clarify the regulatory mechanisms underlying the transcriptomic changes induced by HCV infection and pegIFN-α treatment. In particular, we evaluated the roles of microRNAs (miRNAs) in the regulation of the hepatocellular and immunological host response to HCV infection. Interestingly, we found that the primary transcripts of several miRNAs [including miR-122, which is required for HCV replication (Jopling *et al*, 2005)] are down-regulated following pegIFN-α treatment in the human liver. Consistently, we observe a subtle up-regulation of the corresponding miRNA target genes, indicating that the expression changes observed for the precursor transcripts are reflected in the mature miRNA levels. Although these findings warrant further validation, we propose that the down-regulation of miRNA primary transcripts, in particular for miR-122, may contribute to efficiency of HCV clearance by pegIFN-α/ribavirin treatment.

# Results

## Expression patterns of interferon-stimulated genes define two classes of CHC patients

Previous studies focusing on the response to interferon (IFN) treatment in chronic hepatitis C (CHC) revealed the existence of a subset of patients with high endogenous levels of interferon-stimulated genes (ISGs; Sarasin-Filipowicz *et al*, 2008). This distinction between two categories of CHC patients is highly relevant when seeking to determine the molecular consequences of HCV infection in the human liver, which otherwise can be confounded by the endogenous activation of the IFN system. We thus analyzed the expression levels of ISGs in the examined CHC patients. To do this, we mined a previously published dataset of ISGs (Dill *et al*, 2014) and extracted a set of genes that are strongly up-regulated in the human liver upon pegIFN-α treatment, requiring a minimum expression fold change of 2 across all studied time points. We thus obtained a set of 20 strong ISGs and we assessed their expression levels in 28 liver biopsies (including control non-infected samples, termed hereafter non-CHC) in our dataset (Fig 1A, Dataset EV1).

A hierarchical clustering approach applied on centered and scaled gene expression levels confirmed the existence of two main groups of patients (Fig 1A). The first group, characterized by overall low-ISG transcript levels, comprised 21 samples, including all six non-CHC liver biopsies and 15 of the CHC samples. Importantly, no clear distinction was found between the non-CHC and the CHC samples in this group. The second group, consisting of seven CHC samples, displayed higher expression levels across the 20 analyzed ISGs (Fig 1A). Importantly, we note that this separation between two groups of patients cannot be explained by the HCV genotype carried by the CHC patients, as all four genotypes were found in the seven patients with high-ISG levels (Fig 1A). Similarly, analysis of the inflammation and fibrosis METAVIR scores (Bedossa & Poynard, 1996) and of the HCV viral load indicated that these factors cannot

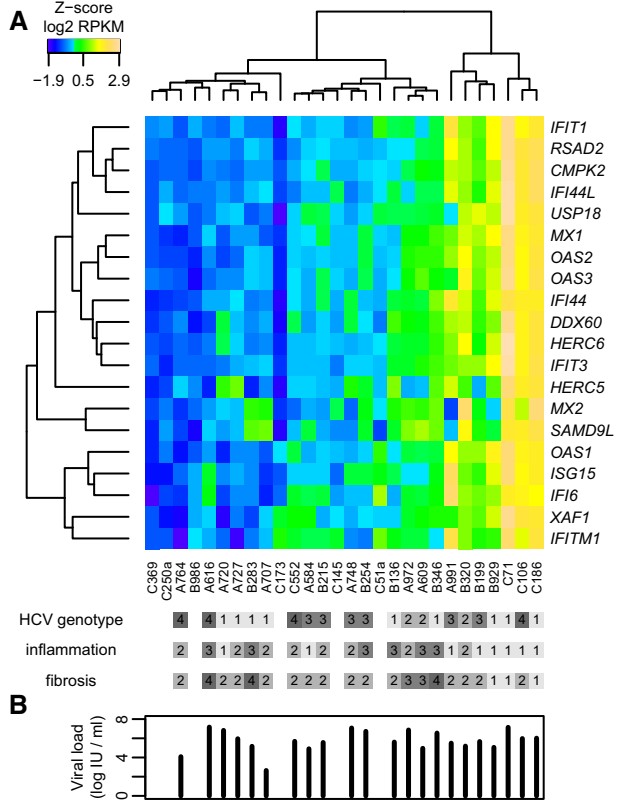

**Figure 1. Endogenous levels of interferon-stimulated levels in human liver.**

A   Expression levels of 20 interferon-stimulated genes (ISGs) in 28 liver biopsies from HCV-infected and non-HCV-infected patients, in the absence of treatment. These ISGs were previously detected as consistently stimulated by pegIFN-α across five time points between 4 and 144 h, in chronic hepatitis C (CHC) patients (Dill et al, 2014). The heatmap represents the Z-score (centered and scaled values) of log2-transformed RPKM (reads per kilobase of exon per million mapped reads) expression levels, normalized based on housekeeping genes (Materials and Methods). The samples (columns) and genes (rows) were hierarchically clustered based on pairwise Euclidean distances. The HCV genotype and the METAVIR scores for inflammation (A0 to A3) and fibrosis (F0 to F4) are displayed for each patient. Blank rectangles correspond to non-infected samples.

B   Barplot of the HCV viral load for each patient, ordered as in (A). Blank spaces correspond to non-infected samples.

explain the patient grouping (Fig 1A and B). To verify that the clustering of CHC samples was not dependent on the set of genes used as markers, we performed a principal component analysis on 360 genes associated with Gene Ontology (GO) categories related to response to interferon (Materials and Methods). This analysis confirmed that our sample sub-classification is robust with respect to the choice of the ISG input dataset (Appendix Fig S1).

**Gene expression changes induced by HCV in the absence of the endogenous IFN system activation**

We first aimed to investigate the gene expression changes induced by HCV infection, without the confounding effect of the activation of the endogenous IFN system, and without the confounding effects

of strong inflammation or fibrosis. We thus compared gene expression levels between six non-CHC and a subset of seven CHC low-ISG samples with METAVIR scores ≤ A2F3, using the Wald test for differential expression implemented in DESeq2 (Love et al, 2014) and a sample randomization procedure to minimize outlier effects (Materials and Methods). We identified 179 robustly differentially expressed protein-coding genes, at a false discovery rate (FDR) threshold of 10% and requiring an absolute fold change above 1.5 (Fig 2A, Dataset EV2). With the same parameters, we discovered 14 long non-coding RNAs (Materials and Methods) and 43 genes with unclear classification (including pseudogenes and other classes of non-coding RNAs; Materials and Methods) that were robustly differentially expressed between the two sample categories (Fig 2A). Most differentially expressed protein-coding genes were up-regulated in the CHC low-ISG patients compared to controls, reaching a maximum fold change of 8.

We next examined the protein-coding genes with the highest absolute fold change between the two groups of samples (Fig 2B). The strongest up-regulated genes included genes typically expressed in immune system cells, including *IGHG1, IGHG3, CD27,* and *CD5* (Fig 2B). Genes specifically associated with defense against viral infections, such as *OASL,* were also strongly up-regulated (Fig 2B). A gene ontology (GO) analysis for up-regulated protein-coding genes revealed strong enrichment for biological processes related to lymphocyte and leukocyte activation, including more specific terms such as T-cell and B-cell activation (Dataset EV3). In contrast, down-regulated genes were enriched for processes related to the protein activation cascade, response to stimulus or complement activation, although these patterns were driven by only a few genes (Dataset EV3).

We then analyzed the expression patterns of these genes in the broad collection of human tissue transcriptomes of the GTEx consortium (Mele et al, 2015). In agreement with the GO association with immune system cells, up-regulated genes were most highly expressed in the whole blood, in the lymphocytes, or in the spleen (Fig 2C), while down-regulated genes generally reached maximum expression in the liver or in adipose tissue (Fig 2C). Moreover, an analysis of the transcription factor binding motifs over-represented in the promoters of the genes up-regulated in low-ISG patients revealed the presence of several transcription factors associated with immune system cells, including members of the ETS family, the E2A, NFKB, and SPIB transcription factors (Appendix Fig S2A). No motif enrichment was found for genes down-regulated in CHC low-ISG samples. Taken together, these results indicate that the gene expression changes observed in this class of HCV-infected patients largely result from the recruitment of immune system cells into the liver.

To further explore the regulatory mechanisms driving differential gene expression patterns, we examined the behavior of microRNA (miRNA) target genes. Experiments in Huh7 hepatocellular carcinoma cells recently showed that HCV functionally sequesters miR-122, thus reducing its binding to endogenous target genes and leading to their up-regulation (Luna et al, 2015). To assess whether this observation also holds *in vivo,* we analyzed the expression fold change of predicted miRNA targets in CHC low-ISG samples compared to non-CHC samples (Fig 2D). We analyzed a set of microRNAs expressed in normal and/or HCV-infected human livers (Hou et al, 2011) and a set of evolutionarily conserved miRNA

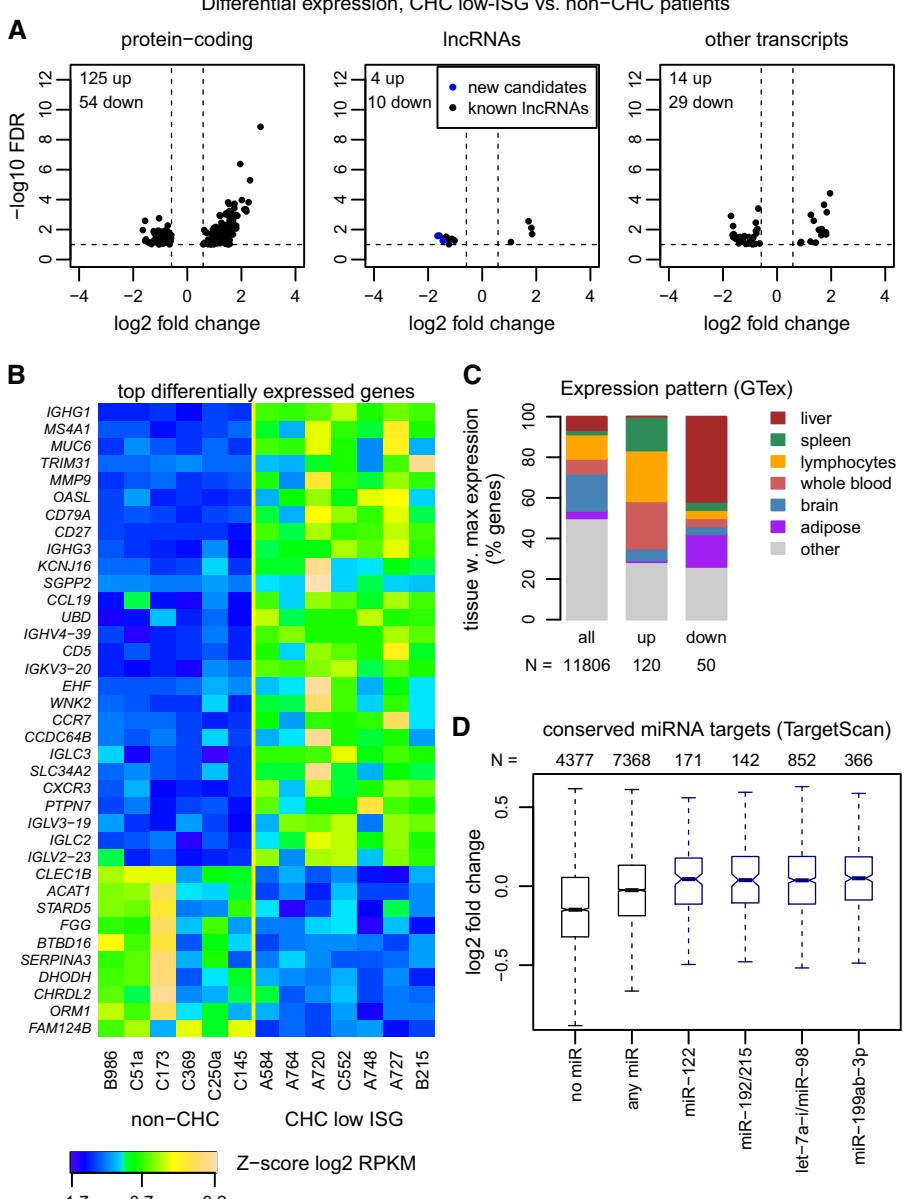

**Figure 2. Differential expression between non-CHC and CHC low-ISG patients.**

A   Volcano plot for the differential expression analysis between non-CHC patients and CHC patients with low levels of endogenous ISG activation (CHC low ISG). The *x*-axis represents the log2-fold expression change in CHC low-ISG patients compared to non-CHC patients. The *y*-axis represents the false discovery rate (with a −log10 transformation) of the differential expression test. Protein-coding genes, candidate long non-coding RNAs (lncRNAs), and other gene categories (including pseudogenes and transcripts with unclear coding potential; Materials and Methods) are represented separately. Genes with false discovery rate (FDR) < 10% and with fold expression change ≥ 1.5 are shown.

B   Heatmap of the expression patterns of the top differentially expressed protein-coding genes. We set a fold change threshold of 3 for up-regulated genes and of 0.5 for down-regulated genes. The heatmap represents the *Z*-score of the log2-transformed RPKM (reads per kilobase of exon per million mapped reads) gene expression levels, normalized based on housekeeping genes (Materials and Methods).

C   Barplots representing the expression patterns of three categories of genes (all expressed genes, genes up-regulated in CHC low-ISG samples, and genes down-regulated in CHC low-ISG samples compared to control non-CHC samples), in the GTEx tissue transcriptome collection. For each expressed gene, we scored the tissue or cell type in which its maximum expression level was reached (Materials and Methods). The height of the rectangles represents the percentage of genes that reaches maximum expression in each of the tissues.

D   Boxplots of the intensity of the expression change (log2-fold) between non-CHC patients and CHC low-ISG patients, for different categories of protein-coding genes defined based on the miRNAs that are predicted to target them. Only miRNAs with high expression in normal or HCV-infected liver were analyzed (Hou *et al*, 2011). Evolutionarily conserved miRNA target predictions were extracted from TargetScan v7.1 (Agarwal *et al*, 2015; Materials and Methods). From left to right: genes that are not conserved targets of any expressed miRNA; genes that are targeted by at least one expressed miRNA; genes that are targeted by miR-122-5p, miR-192/215, let7a-i/miR-98, and miR-199a/199b, respectively. Horizontal bars represent median values. Boxes represent the inter-quartile (25–75%) ranges of the distribution. Boxplot whiskers extend to the most extreme data point found within 1.5 times the inter-quartile distance from the box. Outlier points are not shown.

targets predicted computationally with TargetScan (Agarwal *et al*, 2015; Dataset EV4; Materials and Methods). We found that miR-122 targets had significantly higher fold changes (median 0.045) than targets of other expressed miRNAs (median −0.025, Wilcoxon rank sum test, *P*-value 1e-3) and than genes not targeted by this set of liver miRNAs (median −0.15, Wilcoxon rank sum test, *P*-value < 1e-10; Fig 2D). These observations are compatible with the reported subtle de-repression of miR-122 target genes in the presence of HCV infection (Luna *et al*, 2015). However, we also found that miR-122 targets have comparable expression fold changes with the targets of other highly expressed miRNAs, such as miR-192, let-7, or miR-199 (Fig 2D). Similar conclusions were reached when analyzing a set of miRNA targets identified in Huh7 cells using high-throughput sequencing of RNA isolated by crosslinking immunoprecipitation (Luna *et al*, 2015; Appendix Fig S2B, Dataset EV4). Overall, the fold expression change was positively correlated with the number of distinct miRNA families that are predicted to target each gene (Appendix Fig S2C). This observation cannot be simply explained by the previously reported sequestration of miR-122 by HCV, but may reflect the expression or functional characteristics of genes targeted by multiple miRNA families. Our results thus reveal a potential confounding factor in the up-regulation of miR-122 targets following HCV infection.

## Gene expression patterns associated with endogenous IFN system activation

We next investigated the gene expression changes driven by the combined effect of HCV infection and endogenous IFN system activation. To do this, we contrasted gene expression levels between non-CHC and CHC high-ISG samples (Materials and Methods). Using the same parameters as above, we observed numerous differentially expressed genes in the high-ISG patients, including 503 protein-coding genes, 80 candidate long non-coding RNAs, and 125 other genes (Fig 3A, Dataset EV2). The observed expression fold changes and significance levels spanned a wider range than for the comparison between non-CHC and CHC low-ISG patients (Figs 2A and 3A). As expected, the most highly up-regulated protein-coding genes were known ISGs, including *LAMP3*, *IFI27*, and *RSAD2* (Fig 3B). The up-regulated long non-coding RNAs included a previously described interferon-inducible transcript, *NRIR* (Kambara *et al*, 2014; Fig 3C).

Gene ontology analyses showed a strong enrichment for genes involved in immune system processes, in particular response to virus and type I interferon signaling pathway (Dataset EV3). Interestingly, the GO categories found to be enriched among the genes up-regulated in CHC low-ISG patients were generally also over-represented in this second comparison, although the enrichment was much weaker than the one observed for the interferon pathway (Dataset EV3). In agreement with these observations, we found that interferon-stimulated response element (ISRE) motifs and IFN-regulatory factor (IRF) motifs were strongly over-represented in the promoters of the genes up-regulated in CHC high-ISG samples (Appendix Fig S3). However, we also observed a significant enrichment for NFKB and ERG motifs (Appendix Fig S3), indicating the presence of immune cells in these high-ISG samples. The promoters of down-regulated genes were enriched in binding sites for two liver-specific transcription factors, HNF4a

and HNF1, indicating that most down-regulated genes are hepatocyte-specific genes (Appendix Fig S3). As in the comparison between non-CHC and CHC low-ISG samples (see above), we found no evidence for a specific de-repression of miR-122 target genes (Fig 3D).

Overall, genes differentially expressed between CHC low-ISG and non-CHC samples were recovered in the comparison between CHC high-ISG and non-CHC samples (Fig 4). Specifically, we found that 114 (64%) of the 179 protein-coding genes that differed (up or down) between CHC low-ISG and non-CHC samples were also differentially expressed between CHC high-ISG and non-CHC samples (Fig 4A). The remaining genes that were only differentially expressed in low ISG compared to non-CHC patients generally displayed consistent fold changes in both comparisons, but did not pass the FDR threshold when comparing CHC high-ISG and non-CHC samples (Fig 4B–D). In contrast, most of the genes that were uniquely up-regulated or down-regulated in CHC high-ISG patients compared to controls had only weak expression changes in the comparison between CHC low-ISG and control samples (Fig 4B, E and F). These observations are consistent with the presence of a unique expression signature associated with the group of high-ISG patients (Fig 1A). To further define this expression signature, we directly compared the two groups of CHC samples (Appendix Fig S4, Dataset EV2). We found numerous genes with significant expression changes, including 176 protein-coding genes and 21 lncRNA candidates (Appendix Fig S4). As expected, we observed a strong enrichment for biological processes associated with defense response to virus and type I interferon signaling pathway among the genes up-regulated in high ISG (Dataset EV3).

## Gene expression patterns associated with HCV infection *in vivo* and *in vitro*

We next sought to compare the transcriptional responses following HCV infection *in vivo* and *in vitro*. A previous genome-wide analysis of differential gene expression in HCV-infected Huh-7.5 cells revealed that numerous genes involved in cell death, cell cycle, and cell growth/proliferation are mis-regulated following viral infection (Walters *et al*, 2009). Although our unsupervised gene ontology analyses did not reveal enrichments for cell cycle-associated categories among the genes differentially expressed in CHC samples (Dataset EV3), we found significant intersections between the sets of genes that are differentially expressed following HCV infection *in vivo* and *in vitro*. Specifically, out of 698 protein-coding genes differentially regulated in HCV-infected Huh-7.5 cells (Materials and Methods), we found that 25 (3.6%) were differentially expressed between control and CHC low-ISG samples and 48 (6.9%) were differentially expressed between control and CHC high-ISG samples (FDR < 0.1, Dataset EV5). The extent of the overlap was significantly higher than expected by chance given the total number of differentially expressed genes, in both cases (chi-square test, *P*-value < 1e-9). The common differentially regulated genes included several genes associated with cell death and cell cycle, such as *UBD*, *TNFRSF9, BIRC3, JUN,* and *FOS* (Dataset EV5). Differentially expressed genes shared between HCV-infected Huh-7.5 cells and high-ISG liver biopsies include classical ISGs, such as *MX1, ISG15,* and *IFIT1* (Dataset EV5), as expected given the previously reported

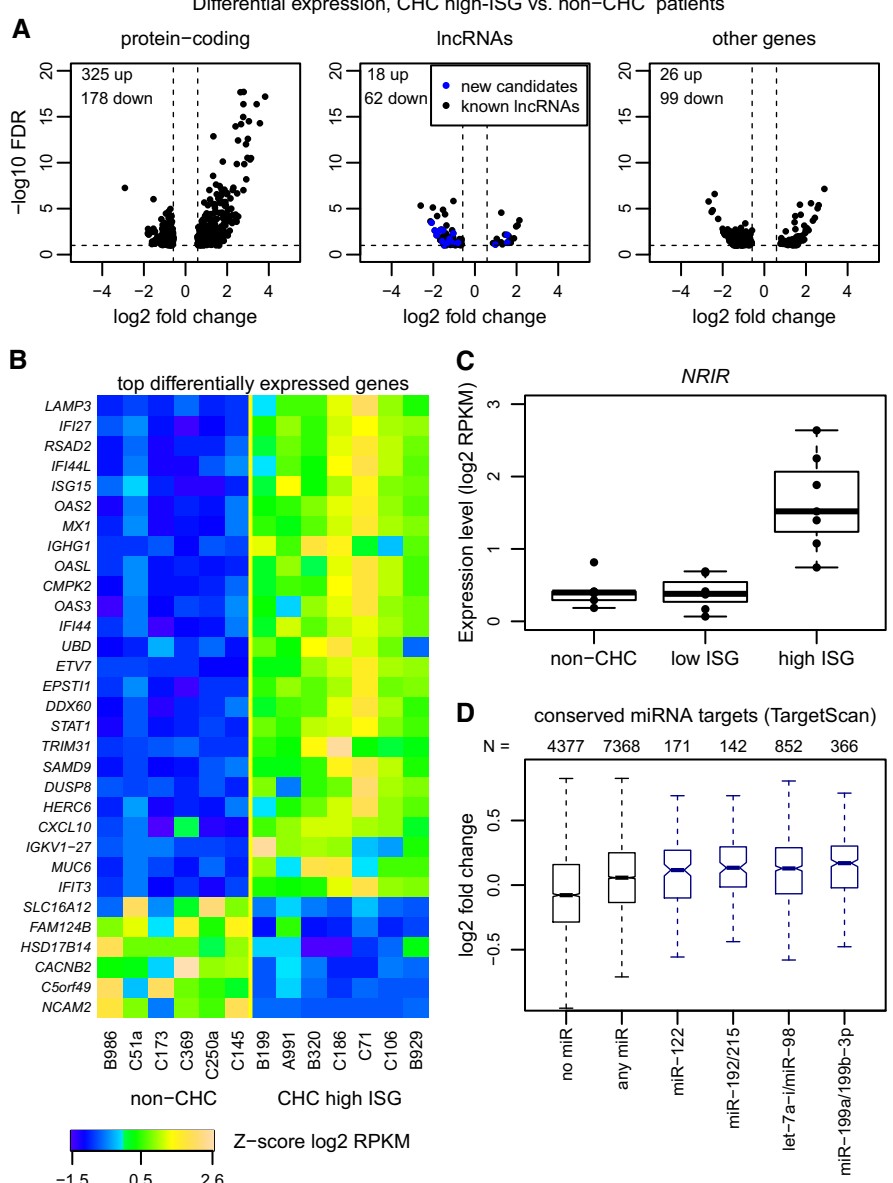

**Figure 3.  Differential expression between non-CHC and CHC high-ISG patients.**

A   Volcano plot for the differential expression analysis between non-CHC patients and CHC patients with high levels of endogenous ISG activation (CHC high ISG). The x-axis represents the log2-fold expression change in CHC high-ISG compared to non-CHC patients. The y-axis represents the false discovery rate (with a −log10 transformation) of the differential expression test. Protein-coding genes, candidate long non-coding RNAs (lncRNAs), and other gene categories (including pseudogenes and transcripts with unclear coding potential; Materials and Methods) are represented separately. Only genes with false discovery rate (FDR) < 10% and with fold expression change > 1.5 are shown.

B   Heatmap of the expression patterns of the top differentially expressed genes between non-CHC and CHC high-ISG patients. To select the top differentially expressed genes, we set a fold change threshold of 5 for up-regulated genes and of 1/3 for down-regulated genes. The heatmap represents the Z-score of the log2-transformed RPKM (reads per kilobase of exon per million mapped reads) gene expression levels, normalized based on housekeeping genes (Materials and Methods).

C   Example of a lncRNA (*NRIR*; Kambara *et al*, 2014) significantly up-regulated in high-ISG patients. The boxplots represent the distribution of expression levels (log2-transformed RPKM) for the three categories of samples. Each point represents an individual sample. Horizontal bars represent median values. Boxes represent the inter-quartile (25–75%) ranges of the distribution. Boxplot whiskers extend to the most extreme data point found within 1.5 times the inter-quartile distance from the box.

D   Boxplots of the intensity of the expression change (log2-fold) between non-CHC patients and CHC high-ISG patients, for different categories of protein-coding genes defined based on the miRNAs that are predicted to target them. Only miRNAs with high expression in normal or HCV-infected liver were analyzed (Hou *et al*, 2011). Evolutionarily conserved miRNA target predictions were extracted from TargetScan v7.1 (Agarwal *et al*, 2015; Materials and Methods). From left to right: genes that are not conserved targets of any expressed miRNA; genes that are targeted by at least one expressed miRNA; genes that are targeted by miR-122-5p, miR-192/215, let7a-i/miR-98, and miR-199a/199b, respectively. Horizontal bars represent median values. Boxes represent the inter-quartile (25–75%) ranges of the distribution. Boxplot whiskers extend to the most extreme data point found within 1.5 times the inter-quartile distance from the box. Outlier points are not shown.

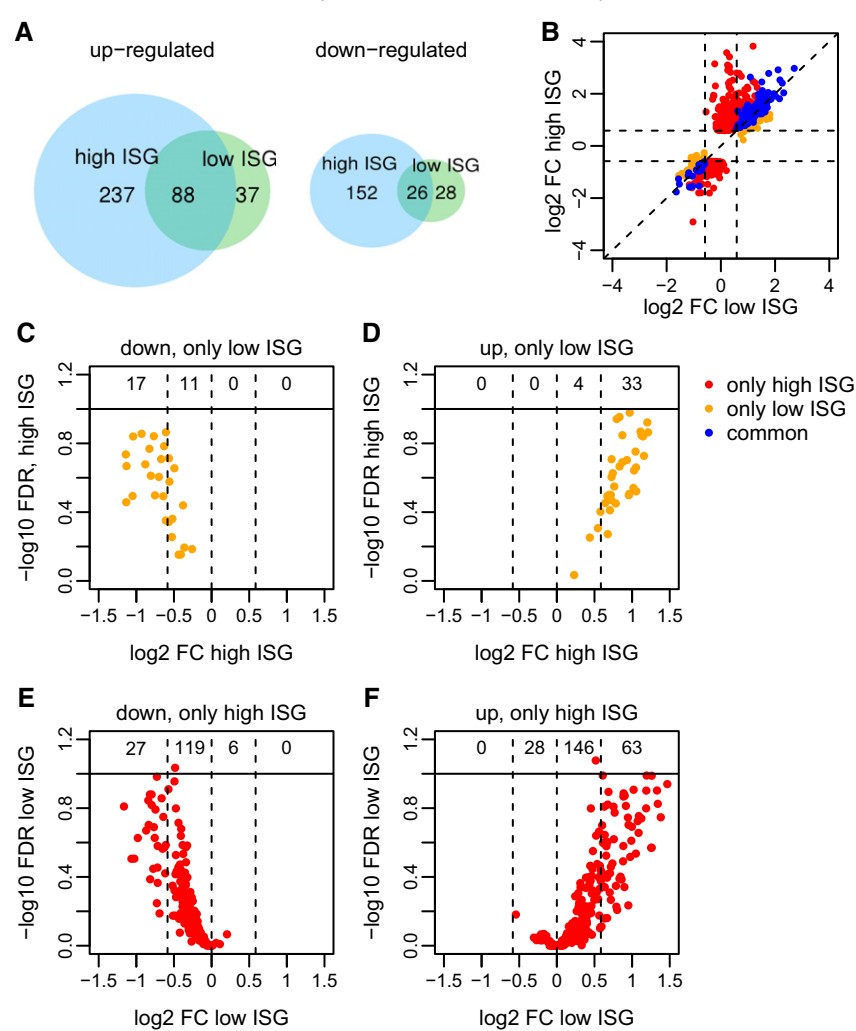

**Figure 4. Differential expression between CHC and non-CHC patients.**

A    Venn diagram depicting the intersection between protein-coding genes that are differentially expressed (FDR < 10%, minimum absolute fold change 1.5) between non-CHC patients, CHC patients with low endogenous ISG levels (CHC low ISG) and CHC patients with high endogenous ISG levels (CHC high ISG). Up-regulated and down-regulated genes are analyzed separately.

B    Comparison of the log2-fold expression change for the two differential expression analyses: x-axis, CHC low-ISG compared with non-CHC patients; y-axis, CHC high-ISG compared with non-CHC patients. Blue: genes significant in both comparisons; red: genes significant only for high-ISG patients; orange: genes significant only for low-ISG patients. Only protein-coding genes are displayed.

C, D    Similar to (B) for genes that are down-regulated (C) or up-regulated (D) only in the comparison between non-CHC and CHC low-ISG patients. The vertical dotted lines represent the absolute fold change threshold of 1.5 (0.58 in log2 scale). The numbers depicted at the top of the plot represent the number of genes in each expression fold change interval (below 1/1.5, between 1/1.5 and 1, between 1 and 1.5, and above 1.5). x-Axis: log2-fold expression change in high-ISG patients compared to controls. y-Axis: log2-fold expression change in low-ISG patients compared to controls.

E, F    Similar to (B) for genes that are down-regulated (E) or up-regulated (F) only in the comparison between non-CHC and CHC high-ISG patients. The vertical dotted lines represent the absolute fold change threshold of 1.5 (0.58 in log2 scale). The numbers depicted at the top of the plot represent the number of genes in each expression fold change interval (below 1/1.5, between 1/1.5 and 1, between 1 and 1.5, and above 1.5). x-Axis: log2-fold expression change in low-ISG patients compared to controls. y-Axis: log2-fold expression change in high-ISG patients compared to controls.

induction of interferon-stimulated genes in these cells (Walters *et al*, 2009).

**Gene expression changes induced by pegIFN-α treatment**

We next analyzed the gene expression changes induced by pegIFN-α/ribavirin treatment in the human liver, by comparing the expression profiles of control and post-treatment biopsies at five different time points ranging from 4 to 144 h post-treatment (Materials and Methods). We considered genes to be differentially expressed if they displayed an absolute fold change of at least 2, at a FDR rate of 5% and an RPKM value above 1 in at least one of the compared samples (Materials and Methods). With these stringent criteria, we observed numerous expression changes at

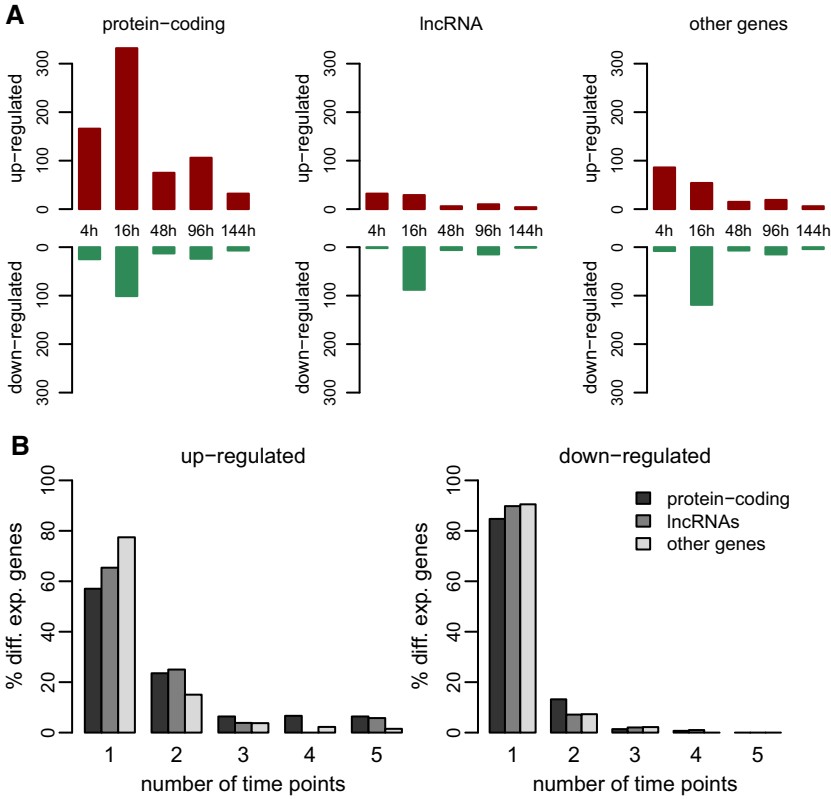

**Figure 5.  Transcriptional response to IFN-α treatment.**

A   Numbers of differentially expressed genes following pegIFN-α/ribavirin treatment in liver biopsies of CHC patients. Gene expression changes were tested between paired biopsies from the same patients, before and after treatment. Protein-coding genes, lncRNAs, and other gene categories (including pseudogenes, and transcripts with unclear coding potential) are displayed separately. We retained genes with a false discovery rate (FDR) < 0.05, a minimum absolute fold change of 2 and expression level (RPKM) > 1 in at least one sample.

B   Barplot representing the proportion of genes that are up-regulated (left) or down-regulated (right) upon pegIFN-α treatment at 1, 2, 3, 4, or 5 time points. Different categories of genes are color-coded.

all time points, in particular for protein-coding genes, but also affecting other categories of transcripts (Fig 5, Dataset EV6). As previously reported (Dill *et al*, 2014), most differentially expressed protein-coding genes were observed at 16 h post-treatment, followed by the 4-h time point (Fig 5A). We observed a different temporal dynamics for differentially expressed non-coding transcripts, for which the number of detected up-regulated genes was highest at the 4-h time point and decreased afterward (Fig 5A). Strikingly, for all gene categories the vast majority of down-regulated genes were observed 16 h after treatment, with only few detected cases elsewhere (Fig 5A).

At all time points, we found strong enrichments in GO categories related to the type I interferon signaling and response to virus pathways among the genes that were up-regulated following pegIFN-α treatment (Dataset EV7), as expected. Consistently, we observed that interferon-stimulated response element (ISRE) motifs and IFN-regulatory factor (IRF) motifs were strongly enriched in the promoters of up-regulated protein-coding genes, at all time points (Appendix Fig S5A–E). For down-regulated genes, we found significant (FDR < 0.1) enrichments in functional categories related to small-molecule biosynthetic process and lipid metabolism, but only for the 16-h time point (Dataset EV7).

We next compared the sets of differentially expressed genes observed for the different time points. We observed that almost 60% of the up-regulated protein-coding genes passed our differential expression thresholds at a single time point (Fig 5B). For up-regulated lncRNAs and other non-coding genes, the time point-specificity was even stronger (Fig 5B). However, we were able to identify a core set of 59 genes that are up-regulated at four or more time points, including 30 genes up-regulated at all analyzed time points (Dataset EV8). Interestingly, three lncRNAs were found to be significantly up-regulated at all time points (Dataset EV8, Appendix Fig S5F–H). However, two of these lncRNAs [including *NRIR*, a lncRNA previously proposed to act as a negative regulator of interferon response (Kambara *et al*, 2014)] are found downstream of interferon-stimulated protein-coding genes (*CMPK2* and *BST2*), and our RNA-seq data suggest that their induction may be at least in part due to leaky transcription from the upstream gene (Appendix Fig S5F and G). No such neighborhood effects were observed for the third lncRNA, which is found upstream of *RHOT1* (Appendix Fig S5H). In contrast to the tendency of up-regulated genes to be shared across time points, down-regulated genes were time point-specific in more than 80% of cases, for all three categories of genes (Fig 5B), and only two genes (including *CD1C* and a newly annotated long

non-coding RNA) were down-regulated at four time points (Dataset EV8).

## Comparison between pegIFN-α treatment and endogenous IFN system activation

The endogenous induction of hundreds of ISGs in patients with CHC has little impact on viral replication, whereas treatment of patients with recombinant pegIFN-α achieves high cure rates specifically in patients without an activation of the endogenous IFN system in the liver (Heim, 2013; Heim & Thimme, 2014). To investigate the molecular differences between these two modes of IFN system activation, we compared the transcriptional response to pegIFN-α treatment with the one elicited by the endogenous IFN system activation. We first extracted the genes that are significantly up- or down-regulated following pegIFN-α treatment, at each time point, and analyzed their expression differences between CHC low-ISG and CHC high-ISG patients (Fig 6, Dataset EV9). We found that the vast majority of genes that are up-regulated upon pegIFN-α treatment are also induced in high-ISG patients (Fig 6, Dataset EV9). In numerous cases, these differences were also statistically significant (FDR < 10%) in the comparison between low-ISG and high-ISG patients (Fig 6). However, the level of gene induction was significantly stronger in the pegIFN-α treatment analysis (Fig 6A), as were the absolute levels of gene expression in the corresponding samples (Fig 6C). In other words, the ISG expression levels reached after pegIFN-α/ribavirin treatment are higher than the ones observed in patients with high endogenous ISG levels. In contrast, genes that were down-regulated upon pegIFN-α treatment were only rarely down-regulated in high ISG compared to low-ISG patients (Fig 6B and D). Similar conclusions were reached when extracting genes that are significantly differentially expressed between low-ISG and high-ISG patients and analyzing their expression patterns following pegIFN-α treatment (Appendix Fig S6). We analyzed the global similarity in expression patterns between high-ISG samples and post-treatment samples, using a principal component analysis and a hierarchical clustering analysis applied to pegIFN-α-affected genes (Appendix Fig S7). Both clustering methods indicated that high-ISG samples are globally similar in expression patterns to the later time points in the pegIFN-α treatment (48, 96 and 144 h). Taken together, these observations suggest that the endogenous IFN system activation and its external stimulation with pegIFN-α/ribavirin treatment have qualitatively similar effects on the gene expression patterns on the human liver transcriptomes. However, numerous quantitative differences in the two transcriptional responses can be observed, with stronger ISG induction levels following pegIFN-α/ribavirin treatment.

To further investigate the inability of the endogenous IFN system activation to cure HCV infections, we analyzed the expression patterns of a set of genes proposed to act as antiviral effectors, selected based on their capacity to inhibit HCV replication in human cell lines (Schoggins et al, 2011; Metz et al, 2012, 2013; Materials and Methods). We analyzed the expression of these genes in our samples (Fig 7, Dataset EV10). We found that their expression levels in pegIFN-α/ribavirin-treated samples and in high-ISG samples were strongly positively correlated (Fig 7A). However, a number of these candidate ISGs were indeed significantly more stimulated by pegIFN-α than by endogenous IFNs in high-ISG patients (Fig 7A, Dataset EV10). In particular, six genes (IRF1, IRF2, IRF7, IRF9, OASL, IFITM3) that were reported as antiviral effectors in at least two publications (Schoggins et al, 2011; Metz et al, 2013) did not differ significantly between low-ISG and high-ISG samples. However, with the exception of IRF2, all of them appeared to be induced in high-ISG patients compared to controls or low-ISG patients, albeit at weak levels. Thus, we could not identify ISGs that are exclusively induced by pegIFN-α and that could be bona fide anti-HCV effector genes.

We extended this analysis to include genes not annotated as antiviral effectors, by extracting all protein-coding genes that were strongly differentially expressed following pegIFN-α/ribavirin treatment (minimum absolute fold change 2, FDR < 0.01, for at least two time points), but which did not differ significantly between low-ISG and high-ISG samples (FDR ≥ 0.1). We found 100 such protein-coding genes, including 85 up-regulated and 15 down-regulated genes (Appendix Fig S8). Note that no genes displayed statistically significant opposite patterns (e.g., up-regulated by pegIFN-α treatment, but down-regulated in high ISG compared to low-ISG samples). This dataset included some genes with known antiviral functions, such as APOBEC3A and OASL, which is potentially relevant for the inability of patients with high endogenous ISG levels to spontaneously clear the viral infection. Overall, a tendency for up-regulation or down-regulation for these genes was also observed in high-ISG samples compared to low-ISG samples, but at lower levels and with more variability among individuals (Appendix Fig S8).

## Down-regulated microRNA host genes following pegIFN-α treatment

Our analysis of the dynamics of differentially expressed genes following pegIFN-α treatment revealed that numerous non-coding transcripts are down-regulated, in particular at the 16-h time point (Fig 5A). We noticed that these down-regulated genes include several miRNA "host" genes (defined as non-coding transcripts that have sense exonic overlap with annotated miRNAs; Materials and Methods). In total, 11 miRNA host genes were significantly differentially expressed (FDR < 0.05) for at least one time point, and all instances were down-regulated rather than up-regulated (Fig 8A). Strikingly, these down-regulated precursors include the "host" gene of miR-122, which enhances HCV replication in human hepatocytes (Jopling et al, 2006; Fig 8A and B). The estimated decrease in expression levels is likely not due to differential processing of the mature miRNA out of the primary transcript, as consistent differences between baseline and post-treatment biopsies were observed along the entire gene length (Fig 8B, Appendix Fig S9). Among the other miRNAs whose precursor genes are down-regulated upon pegIFN-α treatment, miR-146a is a striking example, with significant down-regulation (at 10% FDR) for three out of the five analyzed time points (Fig 8A).

Our RNA-seq dataset does not allow us to determine whether the observed down-regulation of miRNA primary genes affects the pool of mature miRNAs in the cells. We thus assessed the expression levels of mature miRNAs using qPCR, for three miRNAs whose primary transcripts were down-regulated upon pegIFN-α treatment: miR-122-5p, miR-146a-5p, and miR-331-3p (Materials and Methods).

    

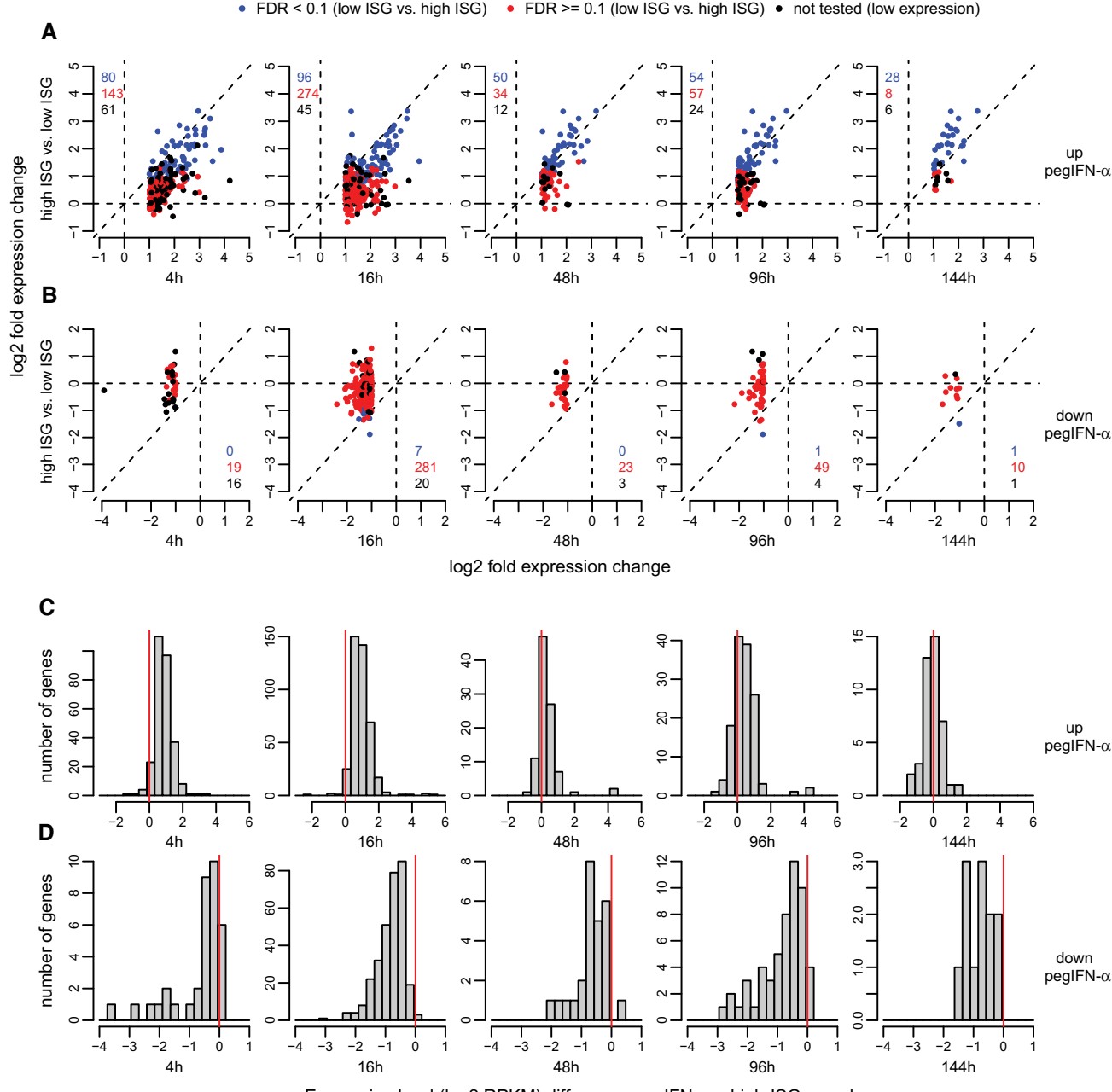

**Figure 6. Common transcriptional signatures of IFN-α treatment and endogenous IFN activity.**

A, B  Differential expression patterns of the genes that are significantly up-regulated (A) or down-regulated (B) (FDR < 0.05 and minimum absolute fold change 2) following pegIFN-α/ribavirin treatment at different time points, in the comparison between CHC low-ISG and CHC high-ISG patients. x-Axis: log2-fold expression change in pegIFN-α-treated compared to control biopsies. y-Axis: log2-fold expression change in high-ISG patients compared to low-ISG patients. Blue dots: genes also significantly differentially expressed in the low ISG versus high-ISG comparison (FDR < 0.1); red dots: genes significant only in the pegIFN-α analysis; black dots: genes not tested in low ISG versus high-ISG comparison due to low or highly variable expression levels (Materials and Methods). The numbers of the genes in each category are depicted on the plot area, with the same color code.

C, D  Histogram of the difference in expression levels (log2-transformed RPKM) between samples treated with pegIFN-α/ribavirin and CHC high-ISG samples, for the genes that are significantly up-regulated (C) or down-regulated (D) (FDR < 0.05 and minimum absolute fold change 2) following pegIFN-α/ribavirin treatment at different time points. The differences were computed between expression levels averaged across all relevant samples.

For all three miRNAs, this data indicates that the mature miRNAs are indeed down-regulated after pegIFN-α treatment at the 16-h time point (also at the 96-h time point for miR-122-5p and miR-146a-5p; Appendix Fig S10, Dataset EV12). To further test this hypothesis

(albeit indirectly), we reasoned that a decrease in mature miRNA expression levels should positively affect the expression of their target genes. We thus analyzed the behavior of predicted miRNA target genes in response to pegIFN-α/ribavirin treatment, at the 16-h

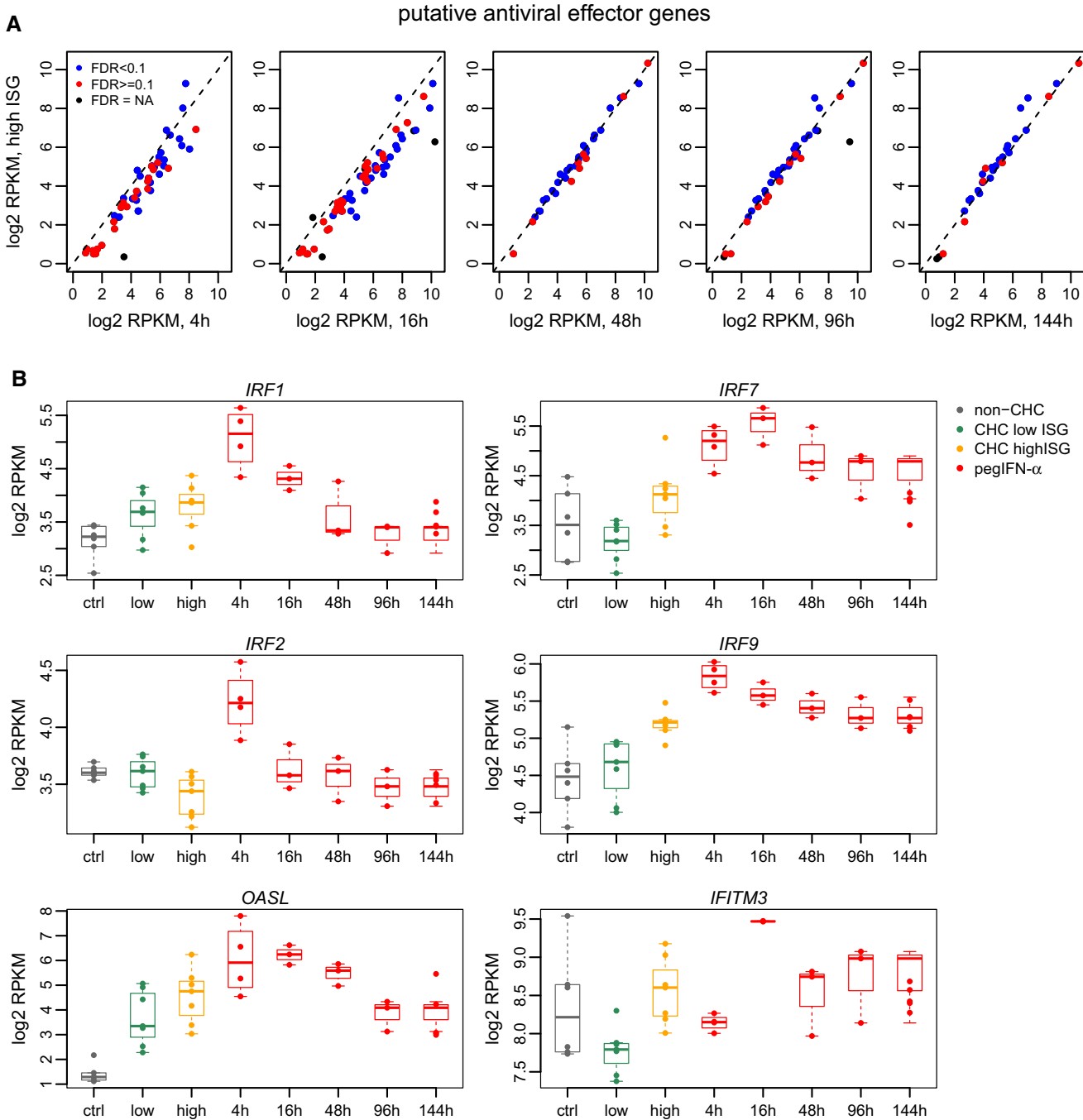

**Figure 7. Regulation of antiviral effectors by IFN-α treatment and endogenous IFN activity.**

A  Comparison between expression levels (log2-transformed RPKM values) in samples obtained after pegIFN-α/ribavirin treatment and in high-ISG patients (in the absence of treatment), for genes predicted to act as antiviral effectors based on experiments in Huh-7.5 cells (Schoggins *et al*, 2011; Metz *et al*, 2013). Only genes that are significantly up-regulated (FDR < 0.05, no fold change threshold) following pegIFN-α/ribavirin treatment are shown. Blue: genes that are also significantly differentially expressed (FDR < 0.1) between low-ISG and high-ISG patients. Red: genes that are not significant in the comparison between low-ISG and high-ISG patients (FDR ≥ 0.1). Black: genes not tested in the comparison between low-ISG and high-ISG patients due to low or variable expression levels (Materials and Methods).

B  Expression patterns of six putative antiviral effector genes that are not significantly different between low-ISG and high-ISG patients (FDR ≥ 0.1). We selected genes reported in both publications (Schoggins *et al*, 2011; Metz *et al*, 2013) which were significantly up-regulated following pegIFN-α/ribavirin treatment in our samples, but not between low-ISG and high-ISG patients. All resulting genes are shown. *y*-Axis: log2-transformed RPKM levels. *x*-Axis: different categories of samples. Gray: control, non-CHC patients; green: CHC low-ISG samples; orange: CHC high-ISG samples; red: biopsies obtained after pegIFN-α/ribavirin treatment, at different time points. The dots represent individual samples. Boxplots are super-imposed over the individual expression levels. Horizontal bars represent median values. Boxes represent the inter-quartile (25–75%) ranges of the distribution. Boxplot whiskers extend to the most extreme data point found within 1.5 times the inter-quartile distance from the box.

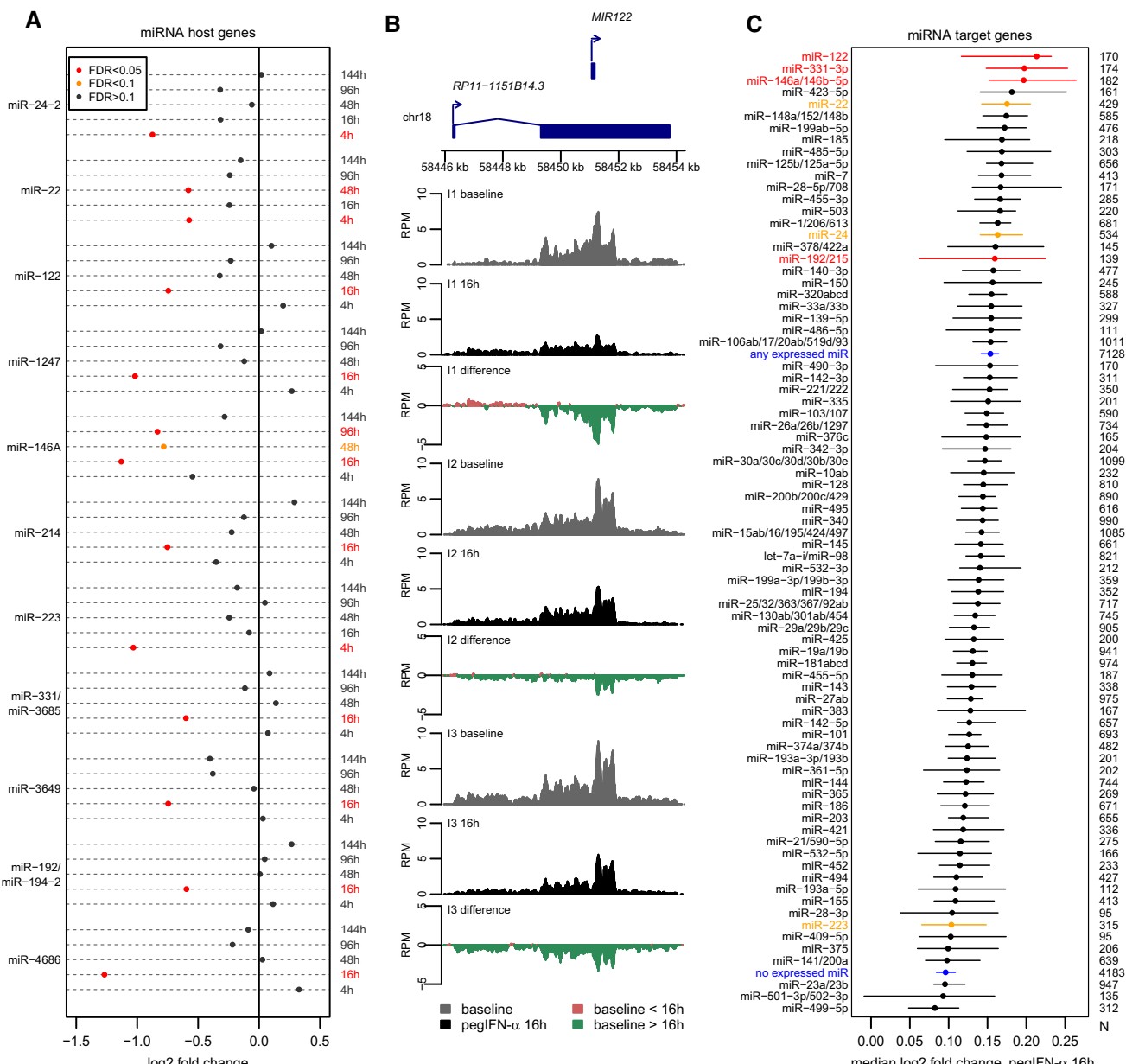

**Figure 8. Down-regulation of miRNA host genes following IFN-α treatment.**

A   Dot-chart representing the expression changes of 11 miRNA host genes following pegIFN-α treatment, at different time points. Red: significant changes, FDR < 0.05; orange: FDR between 0.05 and 0.1; black: FDR > 0.1.

B   RNA-seq coverage profiles along the miR-122 host gene, for pre-treatment and post-treatment biopsies at the 16-h time point, for the three analyzed individuals (here termed I1, I2, and I3). Gray: normalized read coverage for baseline/pre-treatment biopsies. Black: normalized read coverage for post-treatment biopsies. Red: positive difference between post-treatment and pre-treatment biopsies. Green: negative difference between post-treatment and pre-treatment biopsies.

C   Dot-chart of the median fold expression change following pegIFN-α/ribavirin treatment at the 16-h time point, for predicted targets of miRNAs whose hosts are down-regulated (red dots) or of other miRNAs (black dots). We show 79 miRNA families with at least 100 conserved target genes. As a control, we show the median fold expression change for all genes predicted to be targets of any liver-expressed miRNAs or of genes not predicted to be targeted by these miRNAs (blue dots). The horizontal bars represent 95% confidence intervals of the median, constructed with a bootstrap resampling approach (Materials and Methods). Only miRNAs with high expression in normal or HCV-infected liver were analyzed (Hou *et al*, 2011). Evolutionarily conserved miRNA target predictions were extracted from TargetScan v7.1 (Agarwal *et al*, 2015; Materials and Methods). miRNAs highlighted in red correspond to primary miRNA transcripts significantly down-regulated at the 16-h time point; miRNAs highlighted in orange correspond to primary miRNA transcripts significantly down-regulated at other time points.

time point (for which most miRNA host down-regulation events were observed; Fig 8A). As above, we analyzed microRNAs expressed in normal and/or HCV-infected human livers (Hou *et al*,

2011) and their conserved targets predicted computationally with TargetScan (Agarwal *et al*, 2015; Materials and Methods). Strikingly, out of 79 miRNA families with at least 100 conserved target

genes, the three highest median expression fold changes at the 16-h time point were found for miRNAs whose precursor genes were down-regulated following treatment: miR-122, miR-331, and miR-146a/b (Fig 8C). High median fold changes were also observed for miR-22, as well as for miR-24, miR-214, and miR-192 to a lower extent (Fig 8C). We note that this pattern is specific to the 16-h time point; that is, the miRNAs whose primary transcripts are down-regulated do not stand out from the bulk of liver-expressed miRNAs at other time points (Appendix Figs S11 and S12). This result is consistent with our analysis of the temporal dynamics of interferon-induced gene expression, as the strongest changes following pegIFN-α treatment were also observed at the 16-h time point, for most gene categories (Fig 5).

## Discussion

Hepatitis C virus (HCV) is one of the most widely used model systems to investigate host–virus interactions. The adaptive changes to HCV infections have been studied extensively in cell culture systems. For example, a comprehensive analysis of gene expression in HCV-infected Huh-7.5 cells reported over 800 genes with > 2-fold changes in expression (Walters et al, 2009). Adaptive changes were described in lipid biosynthetic pathways, endoplasmatic reticulum stress response, autophagy, and cell cycle regulation. However, due to the difficulties inherent to human liver tissue sampling, few of these findings have been validated in vivo. To investigate whether the observations obtained in cell culture systems can be confirmed in the human liver, we have collected and analyzed transcriptome data from liver biopsies derived from control and chronic hepatitis C patients, in the absence of and during treatment with pegIFN-α/ribavirin.

In the liver, cell-intrinsic adaptive changes to HCV infections are difficult to distinguish from changes induced by the immune response. In experimentally infected chimpanzees, transcriptome analysis revealed a strong induction of hundreds of ISGs in all animals (Bigger et al, 2004). Due to a genetic polymorphism in the IFNL4 gene, ISG induction in humans is variable (Prokunina-Olsson et al, 2013; Terczynska-Dyla et al, 2014). In this study, in order to reduce the strong confounding influence of the endogenous IFN system, we analyzed gene expression patterns separately for patients with or without endogenous IFN activation. We found that gene expression changes between uninfected liver samples and low-ISG samples mainly reflect the presence of immune cell infiltrates in the latter group. However, even in biopsies from patients without ISG induction (low-ISG patients) we could not detect expression changes of genes involved in cellular responses to HCV described in cell culture. For example, a previous large-scale analysis of differential gene expression in HCV-infected Huh-7.5 cells revealed that numerous genes involved in cell death, cell cycle, and cell growth/proliferation are mis-regulated following viral infection (Walters et al, 2009), while our differential expression analyses did not reveal enrichments for these functional categories of genes. We therefore used a targeted approach and specifically analyzed genes previously reported to be changed in HCV-infected Huh-7.5 cells (Walters et al, 2009). Specifically, we could identify a core set of 25 genes that are differentially expressed following HCV infection both in Huh-7.5 cells and in liver samples from patients without IFN system activation. This common gene set included several cell cycle-associated genes, such as UBD, ITIH1, and BIRC3 (Dataset EV5; Walters et al, 2009). Thus, we were able to identify a significant number of genes that respond to HCV infection both in vivo and in vitro.

How might we explain the differences in HCV-induced expression patterns in vivo and in vitro? It has been estimated that the number of HCV virions per infected cell is between 1 and 8 in the human liver, but can be as high as 500–1,000 in the Huh-7.5 cell culture model (Stiffler et al, 2009). It is thus conceivable that some of changes described in cell culture are due to the high viral concentration and do not occur in vivo. However, it is also possible that our analysis was underpowered to detect HCV-induced cell-intrinsic changes in gene expression. Changes in infected hepatocytes could be masked by unchanged gene expression in non-infected hepatocytes and non-parenchymal liver cells (endothelial cells, biliary epithelial cells, stellate cells, fibroblasts, and Kupffer cells). Non-parenchymal cells provide about one-third of the liver mass, and the number of infected hepatocytes can vary from 1 to 55% (Wieland et al, 2014). To minimize the information dilution from uninfected cells, we preferentially included in our study samples from patients with high viral load, because the proportion of infected cells significantly correlates with serum viral load (Wieland et al, 2014). Nevertheless, we cannot exclude that because of these limitations we could not detect some genuine HCV-induced cell-intrinsic changes of gene expression.

We also addressed a long-standing conundrum in the field regarding the inability of the endogenous IFN system activation to eradicate HCV infections. Ever since the discovery that a substantial proportion of patients with CHC have a strong induction of hundreds of ISGs in the liver (Chen et al, 2005; Asselah et al, 2008; Sarasin-Filipowicz et al, 2008) it remained unclear why the endogenous IFN system is ineffective against HCV, whereas therapies with recombinant (pegylated) IFN-α were curative in many patients. Two alternative explanations were discussed: Either some critical ISGs are exclusively induced by pegIFN-α, or pegIFN-α induces the same ISGs but at a higher level than the endogenous IFNs. At first sight, our results indicate that the number of significantly changed ISGs is much higher after pegIFN-α compared to those induced by endogenous IFNs (in high-ISG samples), suggesting that pegIFN-α indeed induces additional ISGs that are not stimulated by endogenous IFNS. However, most of these "additional" genes were up-regulated in high-ISG samples as well, albeit to a lesser degree and therefore not passing the significance threshold. Such quantitative differences were also observed for a number of candidate anti-HCV effector ISGs identified in large screens (Schoggins et al, 2011; Metz et al, 2013). Most of them were significantly stronger induced by pegIFN-α compared to endogenous IFNs, but their expression levels were highly positively correlated between samples treated with pegIFN-α/ribavirin and samples with high endogenous ISG levels (Fig 7, Dataset EV10). We present six candidate antiviral effectors [IRF1, IRF2, IRF7, IRF9, OASL, IFITM3, reported in both publications cited above (Schoggins et al, 2011; Metz et al, 2013)] that do not differ significantly between low-ISG and high-ISG patients. However, with the exception of IRF2, these genes also appear to be slightly induced in high-ISG samples compared to low-ISG or non-infected samples, although at lower levels (Fig 7). Of note, IRF2 stands out because it is induced by pegIFN-α at the 4-h time point, but appears to down-regulated by endogenous IFNs (Fig 7). However, IRF2 is unlikely to

be an antiviral effector *sensu stricto*. IRF2 is a transcriptional regulator involved in IFN induction and in IFN signaling (Ikushima *et al*, 2013). Our *in vivo* analysis therefore could not reliably identify a set of ISG effectors uniquely up-regulated by pegIFN-α that could be responsible for the superior antiviral efficacy of the treatment. Thus, our current analysis for the first time provides strong evidence that quantitative rather than qualitative differences in gene induction are responsible for the failure of the endogenous IFNs and the success of pegIFN-α in viral eradication.

We also addressed the contribution of non-coding RNAs to gene regulation in response to HCV infections and to pegIFN-α treatments. Most prominent of them is miR-122, because it is an essential host factor for HCV replication (Jopling *et al*, 2005). The fascinating role of the highly abundant miR-122 in HCV replication and virus–host interaction has recently been enriched by the observation that binding of miR-122 by HCV can regulate host gene expression by reducing (sponging) the amount of miR-122 available for gene repression (Luna *et al*, 2015). Detailed functional experiments in Huh7 cells brought evidence that HCV infection leads to significant de-repression of miR-122 target genes due to this sponging effect (Luna *et al*, 2015). At first sight, our *in vivo* transcriptomic analysis appeared to confirm this observation, as miR-122 target genes had higher expression levels in HCV-infected compared to control biopsies, more so than genes targeted by all other miRNAs and than genes that are not targets of miRNAs (Figs 2 and 3). However, a closer look revealed the same pattern for other miRNAs highly expressed in the liver (e.g. miR-192, let-7). Because HCV does not bind these other miRNAs, this observation cannot be simply explained by a sponging effect.

An unexpected observation from our analysis of the transcriptional response to pegIFN-α/ribavirin treatment was that 11 long non-coding transcripts that act as precursors for miRNAs (miRNA "host" genes) are significantly down-regulated following treatment. Strikingly, the primary transcript of miR-122 is part of these down-regulated transcripts. Moreover, the genes targeted by these down-regulated miRNAs had the highest median expression fold changes following treatment. We note that only subtle differences were observed for the miRNA target genes, which is suggestive of an expression fine-tuning mechanism, although a miRNA-mediated regulatory process is unlikely to be responsible for major expression changes following pegIFN-α treatment. Nevertheless, this observation indicates that the expression changes observed for the primary transcripts are reflected in the mature miRNA levels, as also indicated by our qPCR analysis for three candidate miRNAs. This finding is in agreement with previous evidence that IFN-beta treatment decreases mature miR-122 levels (Pedersen *et al*, 2007) and that miR-122 levels are lower in patients with endogenous IFN system activation (Sarasin-Filipowicz *et al*, 2009). Interestingly, besides miR-122, several other miRNAs whose host genes are down-regulated have been associated with HCV infections and/or interferon treatment. For example, miR-146a was previously reported to inhibit type I interferon production (Hou *et al*, 2009; Ho *et al*, 2014). Its down-regulation following pegIFN-α/treatment might thus allow for a sustained activation of ISGs and thus more effective antiviral response. Another example is miR-192, which was previously proposed as a predictor for the response to IFN treatment (Motawi *et al*, 2015). Finally, it is tempting to speculate that the down-regulation of miR-122 levels might contribute to the

efficiency of the pegIFN-α/ribavirin treatment in eliminating HCV infections.

In conclusion, this comprehensive gene expression analysis with liver biopsy samples (obtained before and during treatment with pegIFN-α) from patients with HCV infection revealed that HCV has no strong effect on the homeostasis of infected cells, that the endogenous IFN response is qualitatively similar to pegIFN-α treatment but too weak to clear the infection, and that IFN down-regulates miRNA primary transcripts, thereby fine-tuning ISG expression.

# Materials and Methods

## Patient selection

The study included 25 patients with chronic hepatitis C (CHC) and six control patients (not infected with HCV) who underwent a diagnostic liver biopsy in the outpatient clinic of the Division of Gastroenterology and Hepatology, University Hospital Basel. The patients agreed to participate in the study and written informed consent was obtained (approved by the ethics commission of the cantons Basel-Stadt and Basel-Land; approval number EKBBM189/99). All patients with CHC were screened for potential response to treatment using a previously published classification method based on the expression values of *IFI27*, *RSAD2*, *ISG15*, and *HTATIP2* (Dill *et al*, 2011). Patients with high probability of achieving sustained virologic response (SVR) were identified, and in case of planned IFN-based treatment, they were asked to undergo a second liver biopsy. Both biopsies were performed in the morning. Nineteen patients agreed to undergo a second biopsy, as follows: after 4 h (five patients), 16 h (three patients), 48 h (three patients), 96 h (three patients), or 144 h (five patients) of the first therapeutic injection of pegylated interferons (pegIFNs). PegIFNs/Ribavirin doses were set according to HCV genotype and body weight based on standard recommendations. PegIFNs were administered subcutaneously once weekly at the initial dose of 1.5 μg/kg body weight of pegIFN-α-2b (Essex Chemie) or 180 μg of pegIFN-α-2a (Roche). Serum HCV RNA was quantified using the COBAS AmpliPrep/COBAS TaqMan HCV Test and the COBAS Amplicor Monitor from Roche. Diagnosis of control patients was based on clinical, laboratory, and histopathological assessment. For 18 patients, microarray-based expression analyses of the biopsy material were previously published (Dill *et al*, 2014). One additional patient was included for the 16-h time point. Patient characteristics are summarized in Dataset EV1.

## RNA-seq data generation

Total RNA was extracted from fresh-frozen bulk liver biopsy tissue using TRIzol reagent (Invitrogen) and subsequently subjected to DNase treatment using DNA-*free*™ DNA Removal Kit (Ambion) according to manufacturer's instructions. RNA concentration was determined using NanoDrop 2000 spectrophotometer (Thermo Scientific), and RNA quality/integrity was assessed with an Agilent 2100 BioAnalyzer using RNA 6000 Nano Kit (Agilent Technologies). All 46 RNA samples included in this study had RNA integrity number (RIN) values of > 7 (7.1–9.4; median 8.9), with 42 out of 46 samples having RIN values ≥ 8. We generated RNA sequencing (RNA-seq) data using the Illumina TruSeq Stranded mRNA protocol,

with polyA selection. The libraries were sequenced on an Illumina HiSeq 2500 machine, as single-end reads.

### RNA-seq data processing

The RNA-seq reads were trimmed to remove 3′ end adapter sequences, keeping a maximum read length of 81 bp. The reads were aligned on the hg38 primary assembly (excluding patches and haplotypic sequences) of the human genome, downloaded from the Ensembl (Cunningham et al, 2015) database release 76. The alignments were done using TopHat (Kim et al, 2013) release 2.0.10 and Bowtie (Langmead & Salzberg, 2012) release 2.1.0. We allowed intron sizes between 40 bp and 1 million bp for spliced read alignments, with a minimum anchor size of 8 bp and a maximum of one mismatch for each aligned read segment.

### Gene model reconstruction with RNA-seq

We used Cufflinks (Trapnell et al, 2010) release 2.2.1 to reconstruct de novo gene models from TopHat unique read alignments. Reads with multiple reported alignments were excluded from the dataset prior to de novo reconstruction. We kept isoforms with a minimum frequency above 0.05. Intra-intronic transcripts (corresponding to retained introns or unspliced pre-mRNAs) were kept if their frequency was above 0.25. We allowed intron sizes between 40 bp and 500,000 bp. Neighboring transcribed regions were collapsed if they were closer than 40 bp. We performed the gene model reconstruction separately for each RNA-seq sample and then merged them into a single set of gene models using the cuffmerge tool in Cufflinks. We note that de novo reconstructed gene models may be fragmented, meaning that a single locus can be split into several predicted gene models, in particular for low expression levels.

### Long non-coding RNA dataset

We used genomic annotations from the Ensembl (Cunningham et al, 2015) database release 82 as a basis for our analyses, to which we added de novo gene models reconstructed with Cufflinks. We determined the protein-coding potential of de novo gene models based on the codon substitution frequency (CSF) score approach (Lin et al, 2007, 2011) and on sequence similarity with known protein databases [SwissProt (UniProt, 2015)] and protein domains [Pfam-A (Finn et al, 2014)], as previously described (Necsulea et al, 2014). For the CSF approach, to determine the codon substitution frequencies expected for coding and non-coding regions, we aligned Ensembl-annotated protein-coding sequences and intronic regions, for 9,000 1-1 orthologous gene families for 42 vertebrate species extracted from the Ensembl Compara database (Vilella et al, 2009). We then counted all observed codon substitutions and constructed coding and non-coding substitution matrices. We downloaded whole-genome alignments for human and 99 other vertebrate genomes from the UCSC Genome Browser (Rosenbloom et al, 2015) and we computed the CSF score in 75-bp sliding windows along the entire human genome, as described previously (Necsulea et al, 2014). We then extracted all genomic regions with positive CSF scores. As positive CSF scores can appear spuriously on the opposite strand of protein-coding regions (Cabili et al, 2011), for regions with positive CSF scores on both strands we considered only the strand

with the highest score. Gene models were classified as potentially protein-coding if had positive CSF scores over at least 90 bp or 25% of their exonic length. In addition, we searched for sequence similarity between exonic sequences and known proteins from and protein domains from the SwissProt (UniProt, 2015) and Pfam-A (Finn et al, 2014) databases, using blastx (Altschul et al, 1990). We kept SwissProt proteins with high confidence annotations (protein existence score 1, 2, or 3). We retained blastx hits with e-values below 0.001. Gene models were classified as potentially protein-coding if they had significant blastx hits with SwissProt or Pfam-A protein sequences over at least 90 bp or 5% of their exonic length. De novo reconstructed gene models classified as non-coding with both approaches were kept for further long non-coding RNA analyses.

To avoid annotating alternative untranslated regions or introns of protein-coding genes as independent long non-coding RNAs, we further filtered the set of de novo-predicted lncRNAs based on their distance to Ensembl-annotated protein-coding genes. We retained lncRNA candidates that had no sense overlap with Ensembl-annotated protein-coding genes and that were at least 10 kilobases (kb) away from protein-coding gene coordinates. Sense intronic overlaps with other non-coding transcripts were accepted. We discarded loci with an exonic length below 200 bp (for multi-exonic loci) or 500 bp (for mono-exonic loci). We further excluded loci overlapping with RNA repeats or with UCSC-annotated retro-transposed gene copies over more than 10% of their length.

In addition to de novo-predicted lncRNA candidates, we analyzed Ensembl-annotated transcripts corresponding to gene biotypes "lincRNA", "processed_transcript", or "antisense". For both Ensembl-annotated and de novo-predicted lncRNAs, we further required support from at least 50 uniquely mapped RNA-seq reads, across all RNA-seq samples pooled together. In total, we analyzed 8,912 candidate lncRNAs, including 4,246 candidates detected de novo and 4,666 Ensembl-annotated lncRNAs. The de novo annotated lncRNA coordinates are provided with our GEO submission (accession number GSE84346).

### Gene expression estimation

We computed expression levels for protein-coding genes and candidate long non-coding RNAs derived from Ensembl annotations or predicted de novo with RNA-seq. For genes that had multiple isoforms, we combined exon coordinates from all isoforms into a single "flattened" gene model and computed a single expression level for each gene. For protein-coding genes, only protein-coding isoforms were kept, discarding retained introns and other potentially non-functional isoforms. We used two approaches to estimate gene expression levels. First, we estimated gene-based RPKM (reads per kilobase of exon per million mapped reads) values by counting uniquely mapped RNA-seq reads that overlapped with exon coordinates over at least 5 bp. RNA-seq reads that were mapped to sense-overlapping exonic regions were added to the read count of all corresponding genes. The total number of mapped reads for each sample (corresponding to the M denominator in the RPKM computation) was computed after discarding ambiguously mapped reads and reads that aligned to the mitochondrial genome. We normalized expression levels among samples using a previously described median-scaling procedure, based on the least-varying genes in terms of expression ranks (Brawand et al, 2011). Second, we used

Cufflinks to estimate gene expression levels using all TopHat-aligned reads, assigning reads with multiple alignments to each gene depending on gene expression levels estimated with unique reads (default multiple read correction procedure in Cufflinks). All gene expression estimates are provided with our GEO submission (accession number GSE84346).

### Differential gene expression

We assessed differential gene expression using methods in the DESeq2 (Love *et al*, 2014) package (release 1.10.0) in R/Bioconductor (release 3.2.2), starting from the numbers of unambiguous read counts assigned to each gene. To test for differential gene expression following pegIFN-α/ribavirin treatment, we used a likelihood ratio test to compare two generalized linear models: a full model with two explanatory variables (the control/treated condition and the individual) and the reduced model with the individual as a single explanatory variable. To test for differential gene expression between groups of patients (normal liver, CHC high ISG, or CHC low ISG), we compared a model with one explanatory variable (the patient group) and the null model, according to which the patient group has no effect. For the comparison between normal liver, CHC high-ISG and CHC low-ISG patients, we further selected the samples based on their METAVIR score for inflammation and fibrosis (A1/F1, A1/F2, and A2/F2 samples were kept for further analyses). We also excluded one sample (identifier A707), which had high expression levels for inflammatory markers despite its A2/F2 METAVIR classification. The sets of patients analyzed for each test are provided in Dataset EV1. *P*-values were corrected for multiple testing using the Benjamini–Hochberg method, as implemented in DESeq2. Note that for some genes, the resulting false discovery rate (FDR) is set to "NA", if the expression levels are too low or too variable to ensure reliable differential expression estimates (Love *et al*, 2014). For further analyses, we selected genes for which the RPKM level was above 1 in at least one of the compared samples.

### Resampling to control for outlier effects in differential expression analyses

The level of endogenous IFN system activation can vary among patients classified as high ISG (Fig 1). In addition, our dataset included different number of patients in the control (6), low-ISG (7), and high-ISG (7) categories, which may affect the statistical power of the analysis. To avoid outlier effects in our differential expression analyses for these comparisons (contrasting non-CHC, low-ISG, and high-ISG patients), we resampled six out seven patients for the low-ISG and high-ISG categories and we performed differential expression analyses with the reduced number of samples. All possible patient combinations were tested separately. We then selected gene expression differences based on their average false discovery rates (FDR) and average log2-fold expression change level across all resampling replicates. The analyses presented in the manuscript correspond to this resampling correction. Similar conclusions were reached when differential expression analyses were performed with all samples (results provided in Dataset EV2). For these comparisons, we considered that genes are differentially expressed if the FDR was below 0.1 and the absolute fold change above 1.5. We voluntarily chose slightly less restrictive FDR and fold change thresholds for these analyses than for the pegIFN-α treatment analysis, to avoid a loss of sensitivity associated with this resampling procedure.

### Gene ontology enrichment

We performed gene ontology enrichment analyses using the GOrilla webserver (Eden *et al*, 2009). We contrasted a focus set of genes (e.g. genes that were up-regulated in a set of patients or at one time point during treatment) with a background set of genes expressed at in the same samples. We set the false discovery rate for the GO enrichment analysis at 0.1. Only protein-coding genes were used for this analysis. We also downloaded GO associations for each Ensembl-annotated gene from the Ensembl 82 database, using BioMart. For the principal component analysis (PCA) presented in Appendix Fig S1, we selected genes associated with the categories: "type I interferon production", "response to type I interferon", "response to interferon-gamma".

### Transcription factor binding enrichment analysis

We used HOMER (Heinz *et al*, 2010) to assess the enrichment of transcription factor binding motifs in the promoter regions of differentially expressed protein-coding genes, compared to the genomic background. We searched for motifs in a 500-bp region, starting 400 bp upstream of the transcription start site and ending 100 bp downstream. For each analysis, we display the enriched motif, its frequency among the tested genes and the enrichment with respect to the background.

### microRNA expression

To evaluate the involvement of miRNA regulation in the gene expression differences observed between patient categories or following treatment, we first obtained a dataset of miRNAs expressed in the relevant samples. We downloaded miRNA expression values evaluated with RNA sequencing and measured as TPM (tags per million mapped reads) from a previous publication (Hou *et al*, 2011). This dataset included expression levels for normal liver, hepatitis B virus (HBV)-infected and HCV-infected liver, and hepatocellular carcinoma from HBV and HCV-infected livers. For comparability with our samples, we retained only expression levels from normal liver and HCV-infected, non-carcinoma samples. We summed the TPM values across all relevant samples and ranked miRNAs based on their combined expression values. For miRNA target analyses, we retained miRNAs with a combined TPM value of at least 100.

### microRNA target prediction

We used two sources for miRNA target predictions. First, we extracted computational target predictions from the TargetScan v7.1 human dataset (Agarwal *et al*, 2015). To enrich in reliable target predictions, we kept only gene–miRNA associations that had a cumulative context++ score (Agarwal *et al*, 2015) below 0. We analyzed separately miRNA-target gene relationships for which at least one conserved binding site was predicted. Second, we used target predictions determined experimentally with HITS-CLIP in Huh7 cells (Luna *et al*, 2015). This dataset included binding sites

for 50 miRNAs expressed in Huh7 cells. We filtered this dataset to keep only binding sites found in the 3′ UTR region, and we excluded 6mer binding sites. For the TargetScan dataset, we analyzed only miRNAs that were expressed in normal and HCV-infected liver, as defined above. All data are provided in Dataset EV4. For the analysis of miRNA targets and their expression patterns, we always exclusively analyzed genes that are expressed at an RPKM level of at least 1 in at least one of the relevant samples. For the analysis presented in Fig 8, we computed confidence intervals for the median expression fold change (following pegIFN-α/ribavirin treatment at the 16-h time point) with a bootstrap approach, by resampling the same number of genes 100 times with replacement. We then extracted the 2.5 and 97.5% quantiles of the resulting distribution, which are displayed as confidence intervals in Fig 8.

### Definition of microRNA host genes

Our RNA-seq dataset allows us to estimate expression levels for all long, poly-adenylated transcripts, thus in principle including miRNA primary transcripts. We extracted coordinates of miRNA transcripts from Ensembl 82 (Cunningham *et al*, 2015), and we analyzed their overlap with exons of Ensembl-annotated or *de novo*-detected non-coding RNA genes. We found a total of 70 lncRNAs that had exonic overlap with miRNAs, which we classified as potential "miRNA host genes" and included in further analyses (Dataset EV11). We did not include in this analysis genes with intronic overlap with miRNAs.

### Antiviral effector datasets

We specifically analyzed the expression patterns of a subset of ISGs that were previously proposed to act as antiviral effectors through a large-scale over-expression screening approach in Huh-7.5 cells (Schoggins *et al*, 2011). We extracted the list of genes whose over-expression had a negative on HCV replication in this experimental setup, at both 48-h and 72-h time points, and matched them with current Ensembl annotations by gene names. Genes that appeared as having both positive and negative effects on HCV replication (depending on the time points) were excluded. We added to this list of genes a dataset of ISG antiviral effectors previously described (Metz *et al*, 2013), including genes whose antiviral properties were predicted with a knockdown approach (Metz *et al*, 2012). The gene lists are provided in Dataset EV10.

### Differentially expressed genes in HCV-infected Huh7 cells

We also analyzed a set of genes previously shown to be differentially expressed following HCV infection in Huh-7.5 cells (Walters *et al*, 2009). We matched the list of genes with Ensembl 82 annotations using the RefSeq accession number provided in this dataset (Walters *et al*, 2009). The gene lists and their differential expression patterns are provided in Dataset EV5.

### Tissue expression patterns from GTEx

We analyzed the spatial expression patterns of the various gene lists identified in our study, using the gene expression dataset of the

### The paper explained

#### Problem

Hepatitis C virus (HCV) infections are a major cause of morbidity and mortality, which affect an estimated 160 million people worldwide. Despite numerous studies, the molecular impact of HCV infection and of the traditional interferon (IFN)-based treatment on the human liver is not yet fully understood. A subset of HCV patients, which are characterized by high endogenous levels of interferon-stimulated genes, paradoxically do not respond well to interferon therapy. To date, it remains unclear why the endogenous activation of interferon-stimulated genes is unable to clear the virus in these patients.

#### Results

We compared gene expression patterns in the liver for a variety of HCV-infected patients, before and after IFN-based treatment. In patients that do not show an endogenous activity of the IFN system, gene expression patterns predominantly reflect the presence of immune cell infiltrates in the liver. We also investigated the transcriptomes of patients with endogenous IFN activation. We find that the differences in expression patterns between endogenous IFN activity and recombinant IFN therapy are quantitative rather than qualitative. Most IFN-stimulated genes are induced by both recombinant IFN therapy and the endogenous IFN system, but with lower induction levels in the latter. This indicates that the innate immune response in chronic hepatitis C is too weak to clear the virus. Moreover, we found that a specific class of non-coding genes, which generate microRNAs with regulatory roles, are down-regulated following recombinant IFN therapy. These include the primary transcript of miR-122, a host microRNA that is required for HCV infection.

#### Impact

Our results shed light on the gene expression differences between types of HCV-infected patients, which may underlie some patients' inability to respond to interferon therapy. The differential regulation of miR-122 by pegIFN-α/ribavirin therapy may contribute to the efficiency of the treatment.

GTEx consortium, release v6 (Mele *et al*, 2015). We downloaded the pre-computed median RPKM values per tissue from the GTEx server. We used this dataset to estimate the tissue in which each gene reaches its maximum expression, for genes for which the maximum RPKM value in GTEx tissues was at least 1.

### Mature microRNA expression

Mature microRNA expression data were generated starting from the same total RNA preparations that were used for RNA-seq experiments. Equal amount of input total RNA was used in each reaction. RNA was diluted to 4 ng/μl in water and cDNA was generated from 8 ng of total RNA using TaqMan™ Advanced miRNA cDNA Synthesis Kit (Applied Biosystems™ cat# A28007) according to manufacturer's instructions. Briefly, mature miRNAs were first extended on both ends via 3′ polyA tailing and 5′ RNA-adaptor ligation. Next, a reverse transcription (RT) reaction was performed using universal RT primers (supplied with the kit) that recognize the universal sequences present on both the 5′ and 3′ extended ends of the mature miRNAs. One-sixth of RT reaction product was then used to pre-amplify the cDNA using the universal primers (supplied with the kit) that recognize the universal sequences added to all mature miRNAs on the 5′ and 3′ ends. Expression of mature miRNAs miR-122,

miR-146a, and miR-331 was then analyzed by quantitative polymerase chain reaction (qPCR) using TaqMan® Fast Advanced Master Mix (Applied Biosystems™, cat# 4444556) and corresponding TaqMan® Advanced miRNA Assays (Applied Biosystems™, cat# A25576): hsa-miR-122-5p (assay ID: 477855_mir), hsa-miR-146a-5p (assay ID: 478399_mir), and hsa-miR-331-3p (assay ID: 478323_mir). For qPCR, pre-amplified cDNA was diluted 1:10 with water and 5 μl of the diluted product (corresponding to 800 pg of input RNA) was used per reaction. An RT reaction without any template, RT reactions without the reverse transcriptase enzyme and qPCRs without any template served as negative controls. All qPCRs were performed in triplicate.

## Statistics and graphics

All data analyses and graphical representations were done in R. The principal component analyses were performed with functions implemented in the ade4 library.

## Data availability

All raw and processed RNA-seq data, including *de novo* gene annotations obtained with Cufflinks and all expression estimations, are available in the GEO database (accession number GSE84346).

**Expanded View** for this article is available online.

## Acknowledgements
This work was supported by Swiss National Science Foundation (SNF) grants 310030B_147089 and 310030_166202 to M.H. Heim, SNF Ambizione grant PZ00P3_142636 to Anamaria Necsulea, SNF MD-PhD stipend 323530_145255 to Tujana Boldanova, and a SCIEX grant 13.296 to Aleksei Suslov. The computations were performed at the Vital-IT (http://www.vital-it.ch) Center for high-performance computing of the SIB Swiss Institute of Bioinformatics.

## Author contributions
TB recruited patients and performed biopsies. TB and AS extracted RNA and performed qPCR analyses. AN performed bioinformatics analyses. MHH designed and coordinated the study. AN and MHH designed the computational analyses and wrote the manuscript.

## Conflict of interest
The authors declare that they have no conflict of interest.

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
