## [Review Process File · EMBO Molecular Medicine]

Transcriptional response to hepatitis C virus infection and interferon-alpha treatment in the human liver

Tujana Boldanova, Aleksei Suslov, Markus Heim, and Anamaria Necseulea

Corresponding author: Anamaria Necseulea, Ecole Polytechnique Fédérale de Lausanne

Review timeline:

Submission date:	28 August 2016
Editorial Decision:	29 November 2016
Revision received:	12 January 2017
Editorial Decision:	09 February 2017
Revision received:	27 February 2017
Accepted:	02 March 2017

Transaction Report:

Editor: Céline Carret

1st Editorial Decision

29 November 2016

I cannot apologize enough for the impossible delay that occurred with your article due as I told you, to unusual and very unfortunate circumstances. I do realize how frustrated you must be. I am happy to say that we have finally obtained a second review from a referee who accepted to jump in last week.

As whether referee 2 coming back to us remains unclear, we decided to make a decision now, based on 2 reports / 3. Should the missing report be with us within a week or 2, and only should this report suggest realistic improvements to the study but nothing further reaching, will we ask you to address these concerns.

As you will see from the reports below, both referees are enthusiastic about the findings and only have suggestions to make the data more compelling. Given these evaluations, we will consider a revision of your manuscript if you can address the issues that have been raised within the time constraints outlined below. Please note that it is EMBO Molecular Medicine policy to allow only a single round of revision and that, as acceptance or rejection of the manuscript will depend on another round of review, your responses should be as complete as possible.

Revised manuscripts should be submitted within three months of a request for revision; they will otherwise be treated as new submissions, except under exceptional circumstances in which a short

extension is obtained from the editor.

Please read below for important editorial formatting.

I look forward to receiving your revised manuscript.

***** Reviewer's comments *****

Referee #1 (Comments on Novelty/Model System):

I think this can be an important study in an overheated field that has also seen many dodgy, high-impact journal papers. The bioinformatics and statistics are of technically impeccable quality, nothing is overstated or oversold (on the contrary)

Referee #1 (Remarks):

This study by Boldanova et al. addresses an important question in the hepatitis C virus (HCV) field: to what extent does the virus itself - and not secondary effects coming from immune/interferon responses, immune cell infiltration or similar - reprogram hepatic gene expression? An intriguing, additional facet to this issue is that HCV requires the liver-specific miRNA, miR-122, for its replication, a discovery made by Peter Sarnow lab a decade ago (Science 2005) that has led to high interest in HCV-host gene expression interactions ever since. Recent cell culture studies by Bob Darnell's lab (Cell 2015) have led to the suggestion that miR-122 sequestration by HCV derepresses endogenous hepatocyte transcripts and thus directly and supposedly specifically reprograms the host cell transcriptome. Finally, and of clinical importance, it has been known for many years that endogenous interferon responses to HCV in patients can be rather variable and predict the success of interferon treatment therapy (absence of endogenous response leads to better outcomes of IFN treatment). An obvious question is thus also whether endogenous and exogenous IFN lead to qualitatively or quantitatively different gene expression responses that could underlie the observed therapeutic differences. The one difficulty that has hampered progress in answering these seemingly obvious questions is of course the lack of a good HCV animal model and the reliance on human (or, in some cases, primate monkey) material.

The study is based on human material from HCV patients with either high endogenous IFN or such with low endogenous IFN (before and after IFN treatment), plus healthy (non HCV) controls. The last author (Necsulea) has a distinguished bioinformatics background, which shows in an expert and rigorous transcriptomics analysis throughout the manuscript. My prediction is that the datasets will thus serve as an influential reference for HCV researchers - which is one of the important values of the study altogether.

Main findings are that (1) many of the observed gene expression changes (low IFN) stem from immune cell infiltration rather than intrinsic cellular reprogramming; (2) that the gene expression changes engendered by endogenous vs. exogenous/therapeutic IFN are probably mainly of quantitative rather than qualitative nature and that (3) the previously published trans-effect of miR-122 sequestration (see above, Cell 2015) on endogenous mRNAs is not specific for targets of this miRNA and may therefore rather be an unspecific effect altogether. (4) However, the authors propose a specific miR effect that occurs via the transcriptional downregulation of several miR loci.

Points to address:

(1) All concerning Fig 8:

(1A) It would be important to put the analyses shown in Fig. 8 (pri-miRNAs and miRNA targets) on more solid grounds in terms of specificity. First, I feel that it would be important to analyse mature miRNA levels themselves, and not only pri-miRNAs and their predicted targets. This should be done either globally (small RNA-seq / arrays) or specifically for some miRNAs (RT-PCR). In particular, it is difficult to imagine how downregulation of certain pri-miRs at some time-points only (Fig. 8A) can lead to global effects at the target level (Fig. 8C). The mature miRNA levels are thus the link that would be needed to believably connect the two findings.

(1B) The analysis in Fig. 8C uses data from a specific time point (16h) only. It would be interesting

to show the same for the other time points as well, which could give us some idea of the specificity and of whether there is a lasting effect on the miRNA and target population.

(1C) It is unclear to me why some highly expressed miRNAs from Fig. 8A do not show up in Fig. 8C, for example miR-22 and miR-24. They should be added as well.

(2) Fig. 6A: Shouldn't the y-axis read "high ISG vs. low ISG" rather than "low ISG vs. high ISG"?

(3) Also concerning Fig. 6, I was wondering, in general, to which time point of pegIFN α treatment high ISG is most similar in terms of its global gene expression changes? Maybe this could be analysed as it could be interesting to be able to evaluate the two different IFN scenarios and their relationship to each other.

(4) Fig. 7: IRF2 is the only gene that the authors show, whose expression displays the pattern that would be expected from a qualitatively different exogenous vs. endogenous IFN response. Although I understand that the authors restricted this analysis to the known IFN effectors, I think it would be helpful to show all/more genes with a similar response, even if they are not yet known as IFN effectors. This information is in principle already there (one of the Suppl. Tables), but I find it difficult to extract this gene set from the genome-wide tables. Can the authors provide a better table/list with these genes and, depending on how many there are, even a Suppl. Fig. with the corresponding data (similar to the panels in Fig. 7B) ?

(5) Minor point: I was wondering if it would be of interest (and realistic) to include more information on viral load of the patients in the analyses, e.g. to see which gene expression changes scale with viral load.

Referee #3 (Remarks):

This study examines the effects of the endogenous interferon (IFN) and post-treatment pegylated IFN (pegIFN)/ribavirin responses on the overall gene expression profile in liver biopsy samples taken from HCV-infected patients. One interesting conclusion is that, while qualitatively similar, the IFN-stimulated gene expression profile was quantitatively different plus/minus pegIFN treatment. This finding offers an explanation why the endogenous IFN response fails to clear HCV. Another important result was the finding that IFN-response gene expression was provided by infiltrating cells of the immune system and not by cell-intrinsic signaling alterations. This study is perfectly executed using high quality bioinformatics analysis. The description and discussion of the results are very scholarly.

Comments:

Fig. 5: The authors should explain why gene expression changes were only observed at 16hrs after pegIFN/ribavirin treatment.

Fig. 7: Why is IRF2 an outlier in the high ISG group? IRF2 is clearly not the only transcription factor in this group.

Fig. 8: As the authors state, most miR genes are downregulated after pegIFN treatment. However, nearly all of them are upregulated at 144 hrs after treatment. Why is this the case?

As the authors know, pre-miR 122 is modulated in a circadian rhythm in the mouse liver. However, mature miR-122 is very stable, at least in the mouse liver or in cultured human liver cells. Thus, it is imperative to examine the abundance of mature miR-122.

Do the authors know when the biopsies were performed? If they were performed in the morning, as is usually the case, ratios of pre- to mature miR-122s may not reflect steady state abundances (see above).

Referee #1 (Comments on Novelty/Model System):

I think this can be an important study in an overheated field that has also seen many dodgy, high impact journal papers. The bioinformatics and statistics are of technically impeccable quality, nothing is overstated or oversold (on the contrary)

Referee #1 (Remarks):

This study by Boldanova et al. addresses an important question in the hepatitis C virus (HCV) field: to what extent does the virus itself - and not secondary effects coming from immune/interferon responses, immune cell infiltration or similar - reprogram hepatic gene expression? An intriguing, additional facet to this issue is that HCV requires the liver-specific miRNA, miR-122, for its replication, a discovery made by Peter Sarnow lab a decade ago (Science 2005) that has led to high interest in HCV-host gene expression interactions ever since. Recent cell culture studies by Bob Darnell's lab (Cell 2015) have led to the suggestion that miR-122 sequestration by HCV derepresses endogenous hepatocyte transcripts and thus directly and supposedly specifically reprograms the host cell transcriptome. Finally, and of clinical importance, it has been known for many years that endogenous interferon responses to HCV in patients can be rather variable and predict the success of interferon treatment therapy (absence of endogenous response leads to better outcomes of IFN treatment). An obvious question is thus also whether endogenous and exogenous IFN lead to qualitatively or quantitatively different gene expression responses that could underlie the observed therapeutic differences. The one difficulty that has hampered progress in answering these seemingly obvious questions is of course the lack of a good HCV animal model and the reliance on human (or, in some cases, primate monkey) material. The study is based on human material from HCV patients with either high endogenous IFN or such with low endogenous IFN (before and after IFN treatment), plus healthy (non HCV) controls. The last author (Necsulea) has a distinguished bioinformatics background, which shows in an expert and rigorous transcriptomics analysis throughout the manuscript. My prediction is that the datasets will thus serve as an influential reference for HCV researchers - which is one of the important values of the study altogether. Main findings are that (1) many of the observed gene expression changes (low IFN) stem from immune cell infiltration rather than intrinsic cellular reprogramming; (2) that the gene expression changes engendered by endogenous vs. exogenous/therapeutic IFN are probably mainly of quantitative rather than qualitative nature and that (3) the previously published transeffect of miR-122 sequestration (see above, Cell 2015) on endogenous mRNAs is not specific for targets of this miRNA and may therefore rather be an unspecific effect altogether. (4) However, the authors propose a specific miR effect that occurs via the transcriptional downregulation of several miR loci.

Points to address:**(1) All concerning Fig 8:**

(1A) It would be important to put the analyses shown in Fig. 8 (pri-miRNAs and miRNA targets) on more solid grounds in terms of specificity. First, I feel that it would be important to analyse mature miRNA levels themselves, and not only pri-miRNAs and their predicted targets. This should be done either globally (small RNA-seq / arrays) or specifically for some miRNAs (RT-PCR). In particular, it is difficult to imagine how downregulation of certain pri-miRs at some time-points only (Fig. 8A) can lead to global effects at the target level (Fig. 8C). The mature miRNA levels are thus the link that would be needed to believably connect the two findings.

As noted by the reviewer, in figure 8 we show that several primary miRNA transcripts are significantly down-regulated, and that the predicted targets of the respective miRNAs tend to be upregulated following pegIFNa treatment. Nevertheless, we would like to emphasize that the effects observed on the miRNA target genes are subtle, and that we do not claim “global effects at the target level”. As shown in Figure 8C, the targets of the miRNAs whose primary transcripts are downregulated have, as a whole, higher average expression fold changes following pegIFNa treatment than the targets of other miRNAs. However, these differences are small, and we do not claim that a miRNA-mediated regulatory mechanism can be responsible for major expression

changes following pegIFNa treatment. We have further clarified this aspect in the revised text, as follows (discussion, page 17 of the revised manuscript):

“We note that only subtle differences were observed for the miRNA target genes, which is suggestive of an expression fine-tuning mechanism, although a miRNA-mediated regulatory process is unlikely to be responsible for major expression changes following pegIFNa treatment.”

That being said, we agree with both reviewers that data on the mature levels of the miRNAs would provide important support to our observations on primary miRNA transcripts and miRNA target genes. We would like to stress that, in the case of miR-122, previous publications have already shown that its mature miRNA levels are affected by IFN. For example, Pedersen et al. (Nature, 2007), observed a strong down-regulation of miR-122 by IFN beta; lower miR-122 levels were observed in patients that respond poorly to interferon therapy, which generally have high endogenous ISG levels (Sarasin-Filipowicz et al., Nature Medicine, 2009; Tsubota et al., PLoS One, 2014). Thus, the existing literature is in agreement with our observations regarding miR-122.

To directly evaluate this aspect in our study, we selected three miRNAs whose primary transcripts were shown to be down-regulated after pegIFNa treatment (miR-122-5p, miR-146a-5p and miR-331-3p) and quantified their expression levels before and after pegIFNa treatment, by qPCR. For all three miRNAs, this data indicates that the mature miRNAs are indeed down-regulated after pegIFNa treatment at the 16h time-point (also at the 96h time-point for miR-122-5p and miR-146a-5p, Supplementary Figure 10, Supplementary Table 12). Thus, mature miRNA levels indeed appear to be down-regulated by pegIFNa treatment, at least for these three cases. Unfortunately, for several biopsies we did not have enough RNA left to perform qPCR, which makes this analysis considerably less powerful than our RNA-seq analysis of primary miRNA transcripts. For the same reason, we could not evaluate miRNA expression with small RNA-seq or arrays. We have presented this qPCR data in the revised manuscript (pages 13-14 and 26 of the revised text, Supplementary Fig. 10, Supplementary Table 12).

(1B) The analysis in Fig. 8C uses data from a specific time point (16h) only. It would be interesting to show the same for the other time points as well, which could give us some idea of the specificity and of whether there is a lasting effect on the miRNA and target population.

We focused on the 16h time point in Figure 8C because most changes in pri-miRNA transcripts (as well as in other classes of genes) were observed at this time point. Following the reviewer’s suggestion, we have repeated the analysis of miRNA target genes for the other 4 time points, shown in Supplementary Figures 11 and 12. This analysis shows that the effect on miRNA target genes is not found at the other 4 time points, that is, the miRNAs whose primary transcripts are down-regulated do not stand out from the bulk of liver-expressed miRNAs at other time points. This result is consistent with our analysis of the temporal dynamics of interferon-induced gene expression, as the strongest changes following pegIFNa treatment were also observed at the 16h time point, for most gene categories (Figure 5). We have added a comment on this aspect in the text (page 14 in the revised manuscript), as follows:

“We note that this pattern is specific to the 16h time point, that is, the miRNAs whose primary transcripts are down-regulated do not stand out from the bulk of liver-expressed miRNAs at other time points (Supplementary Fig. 11 and 12). This result is consistent with our analysis of the temporal dynamics of interferon-induced gene expression, as the strongest changes following pegIFNa treatment were also observed at the 16h time point, for most gene categories (Fig. 5).”

(1C) It is unclear to me why some highly expressed miRNAs from Fig. 8A do not show up in Fig. 8C, for example miR-22 and miR-24. They should be added as well.

We apologize for this omission. These miRNAs were initially excluded because their identifiers did not perfectly match between the expression dataset (Hou et al., 2011, data from miRBase release 13) and the miRNA target dataset (TargetScan 7.1, miRBase release 21). We have now corrected our procedure for identifying corresponding miRNAs, thus including the highly expressed and conserved miR-24 and miR-22 in Figure 8C. Interestingly, miR-22 is also among the top 5 miRNAs

with the highest median fold expression changes for target genes (Figure 8C), thus adding further support to our initial observations. We note that some miRNAs whose primary transcripts are down-regulated (e.g., miR-1247, miR-223 etc.) cannot be analyzed because they are not part of the set of evolutionarily conserved miRNAs from TargetScan 7.1, which we used for the miRNA target assessment.

(2) Fig. 6A: Shouldn't the y-axis read "high ISG vs. low ISG" rather than "low ISG vs. high ISG"?

We have corrected this mistake.

(3) Also concerning Fig. 6, I was wondering, in general, to which time point of pegIFN α treatment high ISG is most similar in terms of its global gene expression changes? Maybe this could be analysed as it could be interesting to be able to evaluate the two different IFN scenarios and their relationship to each other.

To answer this question, we analyzed the similarity between the expression profiles of high ISG patients and the pegIFN α -treated samples. A principal component analysis and a hierarchical clustering approach based on pairwise Pearson's correlation coefficients among samples (both applied to the set of pegIFN α -stimulated genes) both indicated that high ISG samples are similar to the later time points in the pegIFN α treatment. Indeed, high ISG samples clustered with pegIFN α -treated samples from the 48h, 96h and 144h time points, in both analyses (Figure S7), while the 4h and 16h time points stand out from the other samples. This observation is consistent with the lower induction levels of ISGs observed for the later pegIFN α time points and for the high ISG samples. We have discussed this analysis in the revised manuscript (page 12):

"We analyzed the global similarity in expression patterns between high ISG samples and posttreatment samples, using a principal component analysis and a hierarchical clustering analysis applied to pegIFN α -affected genes (Supplementary Fig. 7). Both clustering methods indicated that high ISG samples are globally similar in expression patterns to the later time points in the pegIFN α treatment (48h, 96h and 144h)."

(4) Fig. 7: IRF2 is the only gene that the authors show, whose expression displays the pattern that would be expected from a qualitatively different exogenous vs. endogenous IFN response. Although I understand that the authors restricted this analysis to the known IFN effectors, I think it would be helpful to show all/more genes with a similar response, even if they are not yet known as IFN effectors. This information is in principle already there (one of the Suppl. Tables), but I find it difficult to extract this gene set from the genome-wide tables. Can the authors provide a better table/list with these genes and, depending on how many there are, even a Suppl. Fig. with the corresponding data (similar to the panels in Fig. 7B) ?

To address this point, we performed a more general analysis of the pegIFN α -affected genes that are not differentially expressed in high ISG patients. We thus selected the protein-coding genes that were strongly up-regulated or down-regulated (minimum expression fold change 2, maximum FDR 0.01) in response to peg IFN α treatment for at least two time points, but which were not differentially expressed in high ISG samples compared to low ISG samples (FDR>0.1). We also searched for genes which showed the opposite expression pattern in high ISG samples (e.g., upregulated by pegIFN α but down-regulated in high ISG samples), but we found none. We found 85 genes up-regulated and 15 genes down-regulated by pegIFN α , which are not differentially expressed between high ISG and low ISG samples. We displayed their expression patterns in Figure S8. These genes include antiviral effectors like *APOBEC3G* and *OASL*, which is potentially relevant for the inability of patients with high endogenous ISG levels to spontaneously clear the viral infection. Overall, figure S8 shows that there is a tendency for up-regulation or down-regulation for these genes in high ISG samples compared to low ISG samples, but at lower levels and with more variability among individuals. Thus, this analysis confirms our initial observation that differences between the endogenous and exogenous IFN response is mainly quantitative rather than qualitative. We have discussed this new analysis in the revised manuscript (page 13):

"We extended this analysis to include genes that were not annotated as antiviral effectors, by extracting all protein-coding genes that were strongly differentially expressed following

pegIFN α /ribavirin treatment (minimum absolute fold change 2, FDR<0.01, for at least two time points), but which did not differ significantly between low ISG and high ISG samples (FDR>=0.1). We found 100 such protein-coding genes, including 85 up-regulated and 15 down-regulated genes (Supplementary Fig. 8). Note that no genes were found significant but with opposite patterns (e.g. up-regulated by pegIFN α treatment, but down-regulated in high ISG compared to low ISG samples). This dataset included some genes with known antiviral functions, such as APOBEC3A and OASL, which is potentially relevant for the inability of patients with high endogenous ISG levels to spontaneously clear the viral infection. Overall, a tendency for up-regulation or down-regulation for these genes was also observed in high ISG samples compared to low ISG samples, but at lower levels and with more variability among individuals (Supplementary Fig. 8)."

(5) Minor point: I was wondering if it would be of interest (and realistic) to include more information on viral load of the patients in the analyses, e.g. to see which gene expression changes scale with viral load.

We display the viral load in Figure 1B. While we agree with the reviewer that this analysis would be relevant, unfortunately our dataset does not include enough patients with low viral loads to address this point.

Referee #3 (Remarks):

This study examines the effects of the endogenous interferon (IFN) and post-treatment pegylated IFN (pegIFN)/ribavirin responses on the overall gene expression profile in liver biopsy samples taking from HCV-infected patients. One interesting conclusion is that, while qualitatively similar, the IFN-stimulated gene expression profile was quantitatively different plus/minus pegIFN treatment. This finding offers an explanation why the endogenous IFN response fails to clear HCV. Another important result was the finding that IFN-response gene expression was provided by infiltrating cells of the immune system and not by cell-intrinsic signaling alterations. This study is perfectly executed using high quality bioinformatics analysis. The description and discussion of the results are very scholarly.

Comments:

Fig. 5: The authors should explain why gene expression changes were only observed at 16hrs after pegIFN/ribavirin treatment.

We have in fact observed gene expression changes at all 5 time points. However, as discussed in our manuscript, the 16h time point does stand out in terms of the numbers of differentially expressed genes, as well as in the intensity of the observed expression changes. This pattern is consistent with previous studies (e.g. Dill et al., Journal of Clinical Investigation, 2014).

Fig. 7: Why is IRF2 an outlier in the high ISG group? IRF2 is clearly not the only transcription factor in this group.

At this point, we cannot propose an explanation for this finding. *IRF2* is the most striking candidate for a qualitative difference between endogenous/exogenous IFN response, among the analyzed set of potential antiviral effectors. In our discussion, we mention that *IRF2* is a transcription factor (page 17 of the revised manuscript) simply to clarify that the analyzed gene set also includes genes with indirect antiviral action, and does not exclusively consist of direct antiviral effectors. However, we do not claim that the *IRF2* pattern can be attributed to its function as a transcription factor.

Fig. 8: As the authors state, most miR genes are downregulated after pegIFN treatment. However, nearly all of them are upregulated at 144 hrs after treatment. Why is this the case?

As mentioned in the main text (page 13 of the revised manuscript), all cases where a primary miRNA transcript was significantly differentially expressed following pegIFN α treatment are instances of down-regulation. All of these cases are shown in Figure 8A. In this figure, we can indeed observe positive (albeit weak) expression fold changes at the 144h time point for 6 out of 11 transcripts, but these are never statistically significant. Thus, we have not discussed this pattern further in the manuscript.

As the authors know, pre-miR 122 is modulated in a circadian rhythm in the mouse liver. However, mature miR-122 is very stable, at least in the mouse liver or in cultured human liver cells. Thus, it is imperative to examine the abundance of mature miR-122.

We agree with both reviewers that an analysis of mature miRNA levels is important to verify our observations. Please see our reply to reviewer 1, which explains in detail the new qPCR analysis performed to address this aspect.

Do the authors know when the biopsies were performed? If they were performed in the morning, as is usually the case, ratios of pre- to mature miR-122s may not reflect steady state abundances (see above).

The biopsies were indeed performed in the morning, for both pre-treatment and post-treatment biopsies. We have clarified this in the methods (page 19 of the revised manuscript). Please note that the pre-treatment samples correspond to baseline biopsies for each patient, performed when the patients first entered the study, while the post-treatment samples were done at controlled time points following the administration of pegIFN α /ribavirin. Because both biopsies were done in the morning, our results regarding the differential expression of miR-122 cannot be confounded by the circadian regulation of pri-miR-122.

2nd Editorial Decision

09 February 2017

Thank you for the submission of your revised manuscript to EMBO Molecular Medicine. We have now received the enclosed reports from the referees that were asked to re-assess it. As you will see the reviewers are now globally supportive and I am pleased to inform you that we will be able to accept your manuscript pending final editorial amendments.

Please submit your revised manuscript within two weeks. I look forward to seeing a revised form of your manuscript as soon as possible.

I look forward to reading a new revised version of your manuscript as soon as possible.

***** Reviewer's comments *****

Referee #1 (Remarks):

This Reviewer is satisfied with the responses to his previous concerns, which have all been addressed, including through additional experiments and analyses.

Referee #3 (Remarks):

This carefully revised manuscript has addressed previously raised concerns.

Corresponding Author Name: Markus H. Heim, Anamaria Necsulea

Manuscript Number: EMM-2016-07006